# Deep Learning for Bayesian Optimization of Scientific Problems with High-Dimensional Structure

**Samuel Kim**                                                              *samkim@mit.edu*
*Department of Electrical Engineering and Computer Science*
*Massachusetts Institute of Technology*

**Peter Y. Lu**
*Department of Physics*
*Massachusetts Institute of Technology*

**Charlotte Loh**
*Department of Electrical Engineering and Computer Science*
*Massachusetts Institute of Technology*

**Jamie Smith**
*Google Research*

**Jasper Snoek**
*Google Research*

**Marin Soljačić**                                                          *soljacic@mit.edu*
*Department of Physics*
*Massachusetts Institute of Technology*

**Reviewed on OpenReview:** *https://openreview.net/forum?id=tPMQ6Je2rB*

## Abstract

Bayesian optimization (BO) is a popular paradigm for global optimization of expensive black-box functions, but there are many domains where the function is not completely a black-box. The data may have some known structure (e.g. symmetries) and/or the data generation process may be a composite process that yields useful intermediate or auxiliary information in addition to the value of the optimization objective. However, surrogate models traditionally employed in BO, such as Gaussian Processes (GPs), scale poorly with dataset size and do not easily accommodate known structure. Instead, we use Bayesian neural networks, a class of scalable and flexible surrogate models with inductive biases, to extend BO to complex, structured problems with high dimensionality. We demonstrate BO on a number of realistic problems in physics and chemistry, including topology optimization of photonic crystal materials using convolutional neural networks, and chemical property optimization of molecules using graph neural networks. On these complex tasks, we show that neural networks often outperform GPs as surrogate models for BO in terms of both sampling efficiency and computational cost.

## 1 Introduction

Bayesian optimization (BO) is a methodology well-suited for global (as opposed to local) optimization of expensive, black-box (e.g. derivative-free) functions and has been successfully applied to a wide range of problems in science and engineering (Ueno et al., 2016; Griffiths & Hernández-Lobato, 2020; Korovina et al., 2020) as well as hyperparameter tuning of machine learning models (Snoek et al., 2012; Swersky et al., 2014; Klein et al., 2017; Turner et al., 2020; Ru et al., 2021). BO works by iteratively deciding the next data

point to label in order to maximize sampling efficiency and minimize the number of data points required to optimize a function, which is critical in many contexts where experiments or simulations can be costly or time-consuming.

However, in many domains, the system is not a complete black box. For example, certain types of high-dimensional input spaces such as images or molecules have some known structure, symmetries and invariances. In addition, the function may be decomposed into other functions; rather than directly outputting the value of the objective, the data collection process may provide intermediate or auxiliary information from which the objective function can be cheaply computed. For example, a scientific experiment or simulation may produce a high-dimensional observation or multiple measurements simultaneously, such as the optical scattering spectrum of a nanoparticle over a range of wavelengths, or multiple quantum chemistry properties of a molecule from a single density functional theory (DFT) calculation. All of these physically-informed insights into the system are potentially useful and important factors for designing surrogate models through inductive biases, but they are often not fully exploited in existing methods and applications.

BO relies on specifying a surrogate model which captures a distribution over potential functions to incorporate uncertainty in its predictions. These surrogate models are typically Gaussian Processes (GPs), as the posterior distribution of GPs can be expressed analytically. However, (1) inference in GPs scales cubically in time with the number of observations and output dimensionality, limiting their use to smaller datasets or to problems with low output dimensionality without the use of kernel approximations, and (2) GPs operate most naturally over continuous low-dimensional input spaces, so kernels for high-dimensional data with complex structure must be carefully formulated by hand for each new domain. Thus, encoding inductive biases can be challenging.

Neural networks (NNs) and Bayesian neural networks (BNNs) have been proposed as an alternative to GPs due to their scalability and flexibility (Snoek et al., 2015; Springenberg et al., 2016). Alternatively, neural networks have also been used to create continuous latent spaces so that BO with vanilla GPs can be more easily applied (Kusner et al., 2017; Tripp et al., 2020). The ability to incorporate a variety of constraints, symmetries, and inductive biases into BNN architectures offers the potential for BO to be applied to more complex tasks with structured data.

This work demonstrates the use of deep learning to enable BO for complex, real-world scientific datasets, without the need for pre-trained models. In particular:

- We take advantage of auxiliary or intermediate information to improve BO for tasks with high-dimensional observations.

- We demonstrate BO on complex input spaces including images and molecules using convolutional and graph neural networks, respectively.

- We apply BO to several realistic scientific datasets, including the optical scattering of a nanoparticle, topology optimization of a photonic crystal material, and chemical property optimization of molecules from the QM9 dataset.

We show that neural networks are often able to significantly outperform GPs as surrogate models on these problems, and we believe that these strong results will also generalize to other contexts and enable BO to be applied to a wider range of problems. We note that while our methods are based on existing methods, we use a novel combination of these methods to tailor existing BO frameworks to real-world, complex applications.

## 1.1 Related Work

Various methods have been formulated to scale GPs to larger problems. For example, Bruinsma et al. (2020) proposes a framework for multi-output GPs that scale linearly with $m$, where $m$ is the dimensionality of a low-dimensional sub-space of the data. Maddox et al. (2021a) uses multi-task GPs to perform BO over problems with large output dimensionalities. Additionally, GPs have been demonstrated on extremely large datasets through the use of GPUs and intelligent preconditioners Gardner et al. (2018); Wang et al. (2019)

or through the use of various approximations Rahimi & Recht (2007); Wang et al. (2018); Liu et al. (2020); Maddox et al. (2021b).

Another approach to scaling BO to larger problems is by combining it with other methods such that the surrogate model does not need to train on the entire dataset. For example, TuRBO uses a collection of independent probabilistic models in different trust regions, iteratively deciding in which trust region to perform BO and thus reducing the problem to a set of local optimizations (Eriksson et al., 2019). Methods such as LA-MCTS build upon TuRBO and dynamically learn the partition function separating different regions (Wang et al., 2020).

GPs have been extended to complex problem settings to enable BO on a wider variety of problems. Astudillo & Frazier (2019) decompose synthetic problems as a composition of other functions, and take advantage of the additional structure to improve BO. However, the multi-output GP they use scales poorly with output dimensionality, and so this approach is limited to simpler problems. This work has also been extended Balandat et al. (2020); Maddox et al. (2021a). GP kernels have also been formulated for complex input spaces including convolutional kernels (Van der Wilk et al., 2017; Novak et al., 2020; Wilson et al., 2016) and graph kernels (Shervashidze et al., 2011; Walker & Glocker, 2019). The graph kernels have been used to apply BO to neural architecture search (NAS) where the architecture and connectivity of a neural network itself can be optimized (Ru et al., 2021).

Deep learning has been used as a scalable and flexible surrogate model for BO. In particular, Snoek et al. (2015) uses neural networks as an adaptive basis function for Bayesian linear regression, which allows BO to scale to large datasets. This approach also enables BO in more complex settings including transfer learning of the adaptive basis across multiple tasks, and modeling of auxiliary signals to improve performance (Perrone et al., 2018). Additionally, Bayesian neural networks (BNNs) that use Hamiltonian Monte Carlo to sample the posterior have been used for single-task and multi-task BO for hyperparameter optimization (Springenberg et al., 2016).

Another popular approach for BO on high-dimensional spaces is latent-space approaches. Here, an autoencoder such as a VAE is trained on a dataset to create a continuous latent space that represents the data. From here, a more conventional optimization algorithm, such as BO using GPs, can be used to optimize over the continuous latent space. This approach has been applied to complex tasks such as arithmetic expression optimization and chemical design (Kusner et al., 2017; Gómez-Bombarelli et al., 2018; Griffiths & Hernández-Lobato, 2020; Tripp et al., 2020; Deshwal & Doppa, 2021). Note that these approaches focus on both data generation and optimization simultaneously, whereas our work focuses on just the optimization process.

Random forests have also been used for iterative optimization such as sequential model-based algorithm configuration (SMAC) as they do not face scaling challenges (Hutter et al., 2011). Tree-structured Parzen Estimators (TPE) are also a popular choice for hyper-parameter tuning (Bergstra et al., 2013). However, these approaches still face the same issues with encoding complex, structured inputs such as images and graphs.

Deep learning has also been applied to improve tasks other than BO. For example, active learning is a similar scheme to BO that, instead of optimizing an objective function, aims to optimize the predictive ability of a model with as few data points as possible. The inductive biases of neural networks has enabled active learning on a variety of high-dimensional data including images (Gal et al., 2017), language (Siddhant & Lipton, 2018), and partial differential equations (Zhang et al., 2019a). BNNs have also been applied to the contextual bandits problem, where the model chooses between discrete actions to maximize expected reward (Blundell et al., 2015; Riquelme et al., 2018).

## 2 Bayesian Optimization

### 2.1 Prerequisites

Now, we briefly introduce the BO methodology; more details can be found in the literature (Brochu et al., 2010; Shahriari et al., 2015; Garnett, 2022). We formulate our optimization task as a maximization problem

in which we wish to find the input $\mathbf{x}^* \in \mathcal{X}$ that maximizes some function $f$ such that $\mathbf{x}^* = \arg\max_{\mathbf{x}} f(\mathbf{x})$. The input $x$ is most simply a real-valued continuous vector, but can be generalized to categorical variables, images, or even discrete objects such as molecules. The function $f$ returns the value of the objective $y = f(x)$ (which we also refer to as the "label" of $x$), and can represent some performance metric that we wish to maximize. In general $f$ can be a noisy function.

A key ingredient in BO is the surrogate model that produces a distribution of predictions as opposed to a single point estimate for the prediction. Such surrogate models are ideally Bayesian models, but in practice, a variety of approximate Bayesian models or even frequentist (i.e. empirical) distributions have been used. In iteration $N$, a Bayesian surrogate model $\mathcal{M}$ is trained on a labeled dataset $\mathcal{D}_{\text{train}} = \{(\mathbf{x}_n, y_n)\}_{n=1}^N$. An acquisition function $\alpha$ then uses $\mathcal{M}$ to suggest the next data point $\mathbf{x}_{N+1} \in \mathcal{X}$ to label, where

$$\mathbf{x}_{N+1} = \arg\max_{\mathbf{x} \in \mathcal{X}} \alpha\left(\mathbf{x}; \mathcal{M}, \mathcal{D}_{\text{train}}\right). \tag{1}$$

The new data is evaluated to get $y_{N+1} = f(\mathbf{x}_{N+1})$, and $(\mathbf{x}_{N+1}, y_{N+1})$ is added to $\mathcal{D}_{\text{train}}$.

## 2.2 Acquisition Function

An important consideration within BO is how to choose the next data point $\mathbf{x}_{N+1} \in \mathcal{X}$ given the model $\mathcal{M}$ and labelled dataset $\mathcal{D}_{\text{train}}$. This is parameterized through the "acquisition function" $\alpha$, which we maximize to get the next data point to label as shown in Equation 1.

We choose the expected improvement (EI) acquisition function $\alpha_{\text{EI}}$ (Jones et al., 1998). When the posterior predictive distribution of the surrogate model $\mathcal{M}$ is a normal distribution $\mathcal{N}(\mu(\mathbf{x}), \sigma^2(\mathbf{x}))$, EI can be expressed analytically as

$$\alpha_{\text{EI}}(\mathbf{x}) = \sigma(\mathbf{x}) \left[ \gamma(\mathbf{x})\Phi(\gamma(\mathbf{x})) + \phi(\gamma(\mathbf{x})) \right], \tag{2}$$

where $\gamma(\mathbf{x}) = (\mu(\mathbf{x}) - y_{\text{best}})/\sigma(\mathbf{x})$, $y_{\text{best}} = \max(\{y_n\}_{n=1}^N)$ is the best value of the objective function so far, and $\phi$ and $\Phi$ are the PDF and CDF of the standard normal $\mathcal{N}(0, 1)$, respectively. For surrogate models that do not give an analytical form for the posterior predictive distribution, we sample from the posterior $N_{\text{MC}}$ times and use a Monte Carlo (MC) approximation of EI:

$$\alpha_{\text{EI-MC}}(\mathbf{x}) = \frac{1}{N_{\text{MC}}} \sum_{i=1}^{N_{\text{MC}}} \max\left(\mu^{(i)}(\mathbf{x}) - y_{\text{best}}, 0\right). \tag{3}$$

where $\mu^{(i)}$ is a prediction sampled from the posterior of $\mathcal{M}$ (Wilson et al., 2018). While works such as Lakshminarayanan et al. (2017) fit the output of the surrogate model to a Gaussian to use Eq. 2 for acquisition, this is not valid when the model prediction for $y$ is not Gaussian, which is generally the case for composite functions (see Section 2.4).

EI has the advantage over other acquisition functions in that the MC approximation (1) remains differentiable to facilitate optimization of the acquisition function in the inner loop (i.e. the MC approximation of upper confidence bound (UCB) is not differentiable and can result in ties) and (2) is inexpensive (i.e. naive Thompson sampling for ensembles would require re-training a model from scratch in each iteration).

## 2.3 Continued Training with Learning Rate Annealing

One challenge is that training a surrogate model on $\mathcal{D}_{\text{train}}$ from scratch in every optimization loop adds a large computational cost that limits the applicability of BO, especially since neural networks are ideally trained for a long time until convergence. To minimize the training time of BNNs in each optimization loop, we use the model that has been trained in the $N$th optimization loop iteration as the initialization (also known as a "warm start") for the $(N + 1)$th iteration, rather than training from a random initialization. In particular, we use the cosine annealing learning rate proposed in Loshchilov & Hutter (2016) which starts with a large learning rate and drops the learning rate to 0. For more details, refer to Section A.3 in the Appendix.

### 2.4 Auxiliary Information

Typically we assume $f$ is a black box function, so we train $\mathcal{M}\colon \mathcal{X} \to \mathcal{Y}$ to model $f$. Here we consider the case where the experiment or observation may provide some intermediate or auxiliary information $\mathbf{z} \in \mathcal{Z}$, such that $f$ can be decomposed as

$$f(\mathbf{x}) = h(g(\mathbf{x})), \tag{4}$$

where $g\colon \mathcal{X} \to \mathcal{Z}$ is the expensive labeling process, and $h\colon \mathcal{Z} \to \mathcal{Y}$ is a known objective function that can be cheaply computed. Note that this is also known as "composite functions" (Astudillo & Frazier, 2019; Balandat et al., 2020). In this case, we train $\mathcal{M}\colon \mathcal{X} \to \mathcal{Z}$ to model $g$, and the approximate EI acquisition function becomes

$$\alpha_{\text{EI-MC-aux}}(\mathbf{x}) = \frac{1}{N_{\text{MC}}} \sum_{i=1}^{N_{\text{MC}}} \max\left( h\left( \mu^{(i)}(\mathbf{x}) \right) - y_{\text{best}}, 0 \right). \tag{5}$$

which can be seen as a Monte Carlo version of the acquisition function presented in Astudillo & Frazier (2019). We denote models trained using auxiliary information with the suffix "-aux." Because $h$ is not necessarily linear, $h\left( u^{(i)}(\mathbf{x}) \right)$ is not in general Gaussian even if $u^{(i)}$ itself may be, which makes the MC approximation convenient or even necessary.

## 3 Surrogate Models

Bayesian models are able to capture uncertainty associated with both the data and the model parameters in the form of probability distributions. To do this, there is a *prior* probability distribution $P(\theta)$ placed upon the model parameters, and the *posterior* belief of the model parameters can be calculated using Bayes' theorem upon observing new data. Fully Bayesian neural networks have been studied in small architectures, but are impractical for realistically-sized neural networks as the nonlinearities between layers render the posterior intractable, thus requiring the use of MCMC methods to sample the posterior. In the last decade, however, there have been numerous proposals for approximate Bayesian neural networks that are able to capture some of the Bayesian properties and produce a predictive probability distribution. In this work, we compare several different options for the BNN surrogate model. In addition, we compare against other non-BNN baselines. We list some of the more notable models here, and model details and results can be found in Section A.4.1 of the Appendix.

**Ensembles** combine multiple models into one model to improve predictive performance by averaging the results of the single models Ensembles of neural networks have been reported to be more robust than other BNNs (Ovadia et al., 2019), and we use "Ensemble" to denote an ensemble of neural networks with identical architectures but different random initializations, which provide enough variation for the individual models to give different predictions. Using the individual models can be interpreted as sampling from a posterior distribution, and so we use Eq. 5 for acquisition. Our ensemble size is $N_{\text{MC}} = 10$.

**Other BNNS**. We also compare to variational BNNs including Bayes by Backprop (BBB) (Blundell et al., 2015) and Multiplicative Normalizing Flows (MNF) (Louizos & Welling, 2017); BOHAMIANN (Springenberg et al., 2016); and NeuralLinear (Snoek et al., 2015). For BBB, we also experiment with KL annealing, denoted by "-Anneal."

**GP Baselines**. GPs are largely defined by their kernel (also called "covariance functions") which determines the prior and posterior distribution, how different data points relate to each other, and the type of data the GP can operate on. In this work, we will use "GP" to refer to a specific, standard specification that uses a Matérn 5/2 kernel, a popular kernel that operates over real-valued continuous spaces. To operate on images, we use a convolutional kernel, labeled as "ConvGP", which is implemented using the infinite-width limit of a convolutional neural network (Novak et al., 2020). Finally, to operate directly on graphs, we use the Weisfeiler-Lehman (WL) kernel as implemented by (Ru et al., 2021), which we label as "GraphGP". The WL kernel is able to operate on undirected graphs containing node and edge features making it appropriate for chemical molecule graphs, and was used by Ru et al. (2021) to optimize neural network architectures in a method they call NAS-BOWL. Additionally, we compare against "GP-aux" which use multi-output GPs for problems with auxiliary information (also known as composite functions) (Astudillo & Frazier, 2019). In

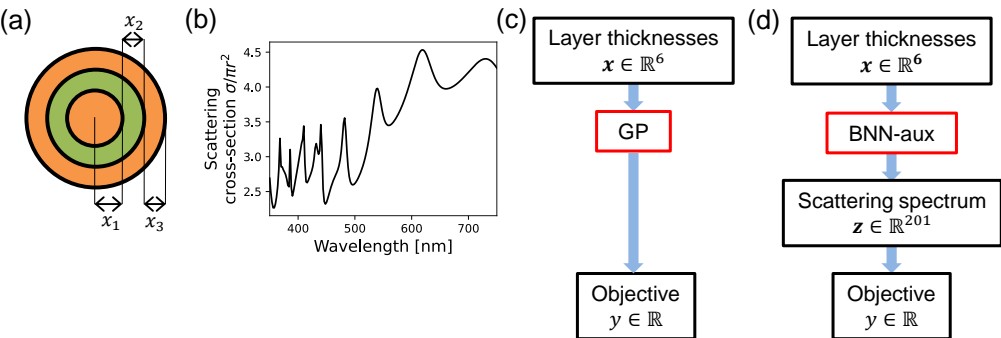

Figure 1: (a) A cross-section of a three-layer nanoparticle parameterized by the layer thicknesses. (b) An example of the scattering cross-section spectrum of a six-layer nanoparticle. (c) Whereas GPs are trained to directly predict the objective function, (d) multi-output BNNs can be trained with auxiliary information, which here is the scattering spectrum.

the Appendix, we also look at GPs that use infinite-width and infinite-ensemble neural network limits as the kernel (Novak et al., 2020) as well as TuRBO which combines GP-based BO with trust regions (Eriksson et al., 2019).

**VAE-GP** uses a VAE trained ahead of time on an unlabelled dataset representative of $\mathcal{X}$. This allows us to encode complex input spaces, such as chemical molecules, into a continuous latent space over which conventional GP-based BO methods can be applied, even enabling generation and discovery of novel molecules that were not contained in the original dataset. Here, we modified the implementation provided by (Tripp et al., 2020) in which they use a junction tree VAE (JTVAE) to encode chemical molecules (Jin et al., 2018). More details can be found in the Appendix.

**Other Baselines**. We compare against two variations of Bayesian optimization, TuRBO (Eriksson et al., 2019) and TPE (Bergstra et al., 2013). We also compare against several global optimization algorithms that do not use surrogate models and are cheap to run, including LIPO (Malherbe & Vayatis, 2017), DIRECT-L (Gablonsky & Kelley, 2001), and CMA-ES.

We emphasize that ensembles and variational methods can easily scale up to high-dimensional outputs with minimal increase in computational cost by simply changing the output layer size. Neural Linear and GPs scale cubically with output dimensionality (without the use of covariance approximations), making them difficult to train on high-dimensional auxiliary or intermediate information.

## 4  Results

We now look at three real-world scientific optimization tasks all of which provide intermediate or auxiliary information that can be leveraged. In the latter two tasks, the structure of the data also becomes important and hence BNNs with various inductive biases significantly outperform GPs and other baselines. For simplicity, we only highlight results from select architectures (see Appendix for full results along with dataset and hyperparameter details). All BO results are averaged over multiple trials, and the shaded area in the plots represents ± one standard error over the trials.

### 4.1  Multilayer Nanoparticle

We first consider the simple problem of light scattering from a multilayer nanoparticle, which has a wide variety of applications that demand a tailored optical response (Ghosh Chaudhuri & Paria, 2012) including biological imaging (Saltsberger et al., 2012), improved solar cell efficiency (Ho et al., 2012; Shi et al., 2013), and catalytic materials (Tang & Henkelman, 2009). In particular, the nanoparticle we consider consists of a lossless silica core and 5 spherical shells of alternating $TiO_2$ and silica. The nanoparticle is parameterized by the core radius and layer thicknesses as shown in Figure 1(a), which we restrict to the range 30 nm to

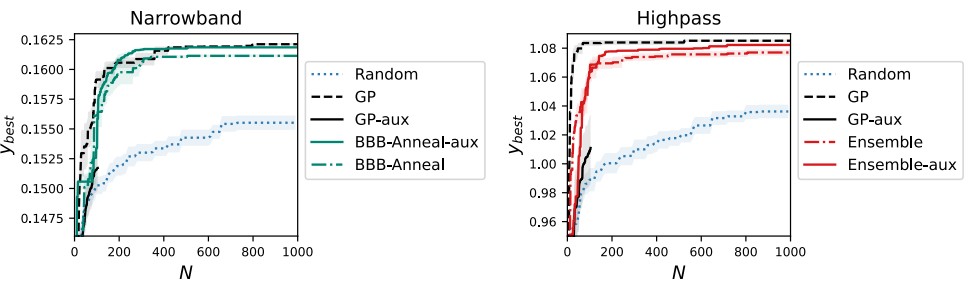

Figure 2: BO results for two different objective functions for the nanoparticle scattering problem. Training with auxiliary information (where $\mathcal{M}$ is trained to predict $\mathbf{z}$) is denoted with "-aux". Adding auxiliary information to BNNs significantly improves performance.

70 nm. Because the size of the nanoparticle is on the order of the wavelength of light, its optical properties can be tuned by the number and thicknesses of the layers. The scattering spectrum can be calculated semi-analytically, as detailed in Section A.1.1 of the Appendix.

We wish to optimize the scattering cross-section spectrum over a range of visible wavelengths, an example of which is shown in Figure 1(b). In particular, we compare two different objective functions: the narrowband objective that aims to maximize scattering in the small wavelength range 600 nm to 640 nm and minimize it elsewhere, and the highpass objective that aims to maximize scattering above 600 nm and minimize it elsewhere. While conventional GPs train using the objective function as the label directly, BNNs with auxiliary information can be trained to predict the full scattering spectrum, i.e. the auxiliary information $\mathbf{z} \in \mathbb{R}^{201}$, which is then used to calculate the objective function, as shown in Figure 1(c,d).

BO results are shown in Figure 2. Adding auxiliary information significantly improves BO performance for BNNs. Additionally, they are competitive with GPs, making BNNs a viable approach for scaling BO to large datasets. In Appendix A.5, we see similar trends for other types of BNNs. Due to poor scaling of multi-output GPs with respect to output dimensionality, we are only able to run GP-AUX for a small number of iterations in a reasonable time. Within these few iterations, GP-AUX performs poorly, only slightly better than random sampling. We also see in the Appendix that BO with either GPs or BNNs are comparable with, or outperform other global optimization algorithms, including DIRECT-L and CMA-ES.

## 4.2 Photonic Crystal Topology

Next we look at a more complex, high-dimensional domain that contains symmetries not easily exploitable by GPs. Photonic crystals (PCs) are nanostructured materials that are engineered to exhibit exotic optical properties not found in bulk materials, including photonic band gaps, negative index of refraction, and angular selective transparency (John, 1987; Yablonovitch, 1987; Joannopoulos et al., 2008; Shen et al., 2014). As advanced fabrication techniques are enabling smaller and smaller feature sizes, there has been growing interest in inverse design and topology optimization to design even more sophisticated PCs (Jensen & Sigmund, 2011; Men et al., 2014) for applications in photonic integrated circuits, flat lenses, and sensors (Piggott et al., 2015; Lin et al., 2019).

Here we consider 2D PCs consisting of periodic unit cells represented by a $32 \times 32$ pixel image, as shown in Figure 3(a), with white and black regions representing vacuum (or air) and silicon, respectively. Because optimizing over raw pixel values may lead to pixel-sized features or intermediate pixel values that cannot be fabricated, we have parameterized the PCs with a level-set function $\phi \colon \mathcal{X} \to \mathcal{V}$ that converts a 51-dimensional feature vector $\mathbf{x} = [c_1, c_2, ..., c_{50}, \Delta] \in \mathbb{R}^{51}$ representing the level-set parameters into an image $\mathbf{v} \in \mathbb{R}^{32 \times 32}$ that represents the PC. More details can be found in Section A.1.2 in the Appendix.

We test BO on two different data distributions, which are shown in Figure 3(b,c). In the PC-A distribution, $\mathbf{x}$ spans $c_i \in [-1, 1]$, $\Delta \in [-3, 3]$. In the PC-B distribution, we arbitrarily restrict the domain to $c_i \in [0, 1]$. The PC-A data distribution is translation invariant, meaning that any PC with a translational shift will also

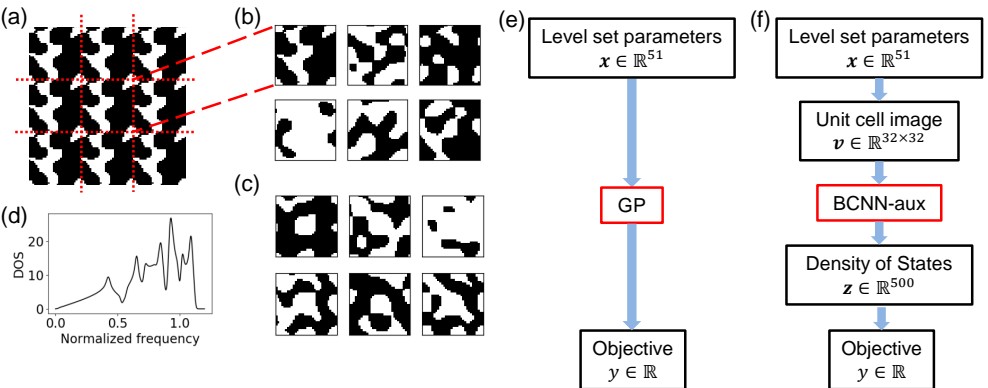

Figure 3: (a) A 2D photonic crystal (PC). The black and white regions represent different materials, and the periodic unit cells are outlined in red. Examples of PC unit cells drawn from the (b) PC-A distribution and (c) the PC-B distributions. The PC-A data distribution is translation invariant, whereas unit cells drawn from the PC-B distribution all have white regions in the middle of the unit cell, so the distribution is not translation invariant. (d) Example of a PC density of states (DOS). (e, f) Comparison of the process flow for training the surrogate model in the case of (e) GPs and (f) Bayesian Convolutional NNs (BCNN). The BCNN can train directly on the images to take advantage of the structure and symmetries in the data, and predict the multi-dimensional DOS.

be in the data distribution. However, the PC-B data distribution is not translation invariant, as shown by the white regions in the center of all the examples in Figure 3(c).

The optical properties of PCs can be characterized by their photonic density of states (DOS), e.g. see Figure 3(d). We choose an objective function that aims to minimize the DOS in a certain frequency range while maximizing it everywhere else, which corresponds to opening up a photonic band gap in said frequency range. As shown in Figure 3(e,f), we train GPs directly on the level-set parameters $\mathcal{X}$, whereas we train the Bayesian convolutional NNs (BCNNs) on the more natural unit cell image space $\mathcal{V}$. BCNNs can also be trained to predict the full DOS as auxiliary information $\mathbf{z} \in \mathbb{R}^{500}$.

The BO results, seen in Figure 4(a), show that BCNNs outperform GPs by a significant margin on both datasets, which is due to both the auxiliary information and the inductive bias of the convolutional layers, as shown in Figure 4(b). Because the behavior of PCs is determined by their topology rather than individual pixel values or level-set parameters, BCNNs are much better suited to analyze this dataset compared to GPs. Additionally, BCNNs can be made much more data-efficient since they directly encode translation invariance and thus learn the behavior of a whole class of translated images from a single image. Because GP-AUX is extremely expensive compared to GP ($500\times$ longer on this dataset), we are only able to run GP-AUX for a small number of iterations, where it performs comparably to random sampling. We also compare to GPs using a convolutional kernel ("ConvGP-NNGP") in Figure 4(a). ConvGP-NNGP only performs slightly better than random sampling, which is likely due to a lack of auxiliary information and inflexibility to learn the most suitable representation for this dataset.

For our main experiments with BCNNs, we use an architecture that respects translation invariance. To demonstrate the effect of another commonly used deep learning training technique, we also experiment with incorporating translation invariance into a translation dependent (i.e. *not* translation invariant) architecture using a data augmentation scheme in which each image is randomly translated, flipped, and rotated during training. We expect data augmentation to improve performance when the data distribution exhibits the corresponding symmetries: in this case, we focus on translation invariance. As shown in Figure 4(c), we indeed find that data augmentation improves the BO performance of the translation dependent architecture when trained on the translation invariant PC-A dataset, even matching the performance of a translation invariant architecture on PC-A. However, on the translation dependent PC-B dataset, data augmentation initially hurts the BO performance of the translation dependent architecture because the model is unable to

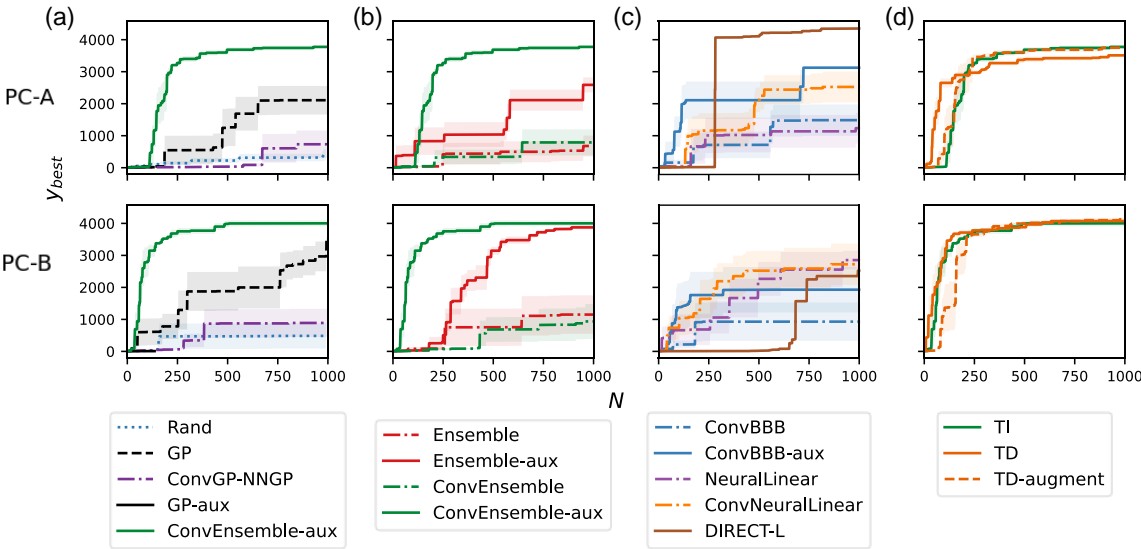

Figure 4: Three sets of comparisons for BO results on the (top row) PC-A and (bottom row) PC-B datasets. (a) BNNs with inductive biases outperform all other GP baselines and the random baseline. Note that GP-AUX is comparable to random sampling. (b) The inductive bias of convolutional layers and the addition of auxiliary information significantly improve performance of BCNNs. (c) Additional comparisons. (d) Data augmentation boosts performance if the augmentations reflect a symmetry present in the dataset but not enforced by the model architecture. "TI" refers to a translation invariant BCNN architecture, whereas "TD" refers to a translation dependent architecture. "-augment" signifies that data augmentation of the photonic crystal image is applied, which includes periodic translations, flips, and rotations.

quickly specialize to the more compact distribution of PC-B, putting its BO performance more on par with models trained on PC-A. These results show that techniques used to improve generalization performance (such as data augmentation or invariant architectures) for training deep learning architectures can also be applied to BO surrogate models and, when used appropriately, directly translate into improved BO performance. Note that data augmentation would not be feasible for GPs without a hand-crafted kernel as the increased size of the dataset would cause inference to become computationally intractable.

### 4.3 Organic Molecule Quantum Chemistry

Finally, we optimize the chemical properties of molecules. Chemical optimization is of significant interest in both academia and industry with applications in drug design and materials optimization (Hughes et al., 2011). This is a difficult problem where computational approaches such as density functional theory (DFT) can take days for simple molecules and are intractable for larger molecules; synthesis is expensive and time-consuming, and the space of synthesizable molecules is large and complex. There have been many approaches for molecular optimization that largely revolve around finding a continuous latent space of molecules (Gómez-Bombarelli et al., 2018) or hand-crafting kernels to operate on molecules (Korovina et al., 2020).

Here we focus on the QM9 dataset (Ruddigkeit et al., 2012; Ramakrishnan et al., 2014), which consists of 133,885 small organic molecules along with their geometric, electronic, and thermodynamics quantities that have been calculated with DFT. Instead of optimizing over a continuous space, we draw from the fixed pool of available molecules and iteratively select the next molecule to add to $\mathcal{D}_{\text{train}}$. This is a problem setting especially common to materials design where databases are incomplete and the space of experimentally-feasible materials is small.

We use a Bayesian graph neural network (BGNN) for our surrogate model, as GNNs have become popular for chemistry applications due to the natural encoding of a molecule as a graph with atoms and bonds as nodes and edges, respectively. For baselines that operate over continuous spaces (i.e. GPs and simple neural

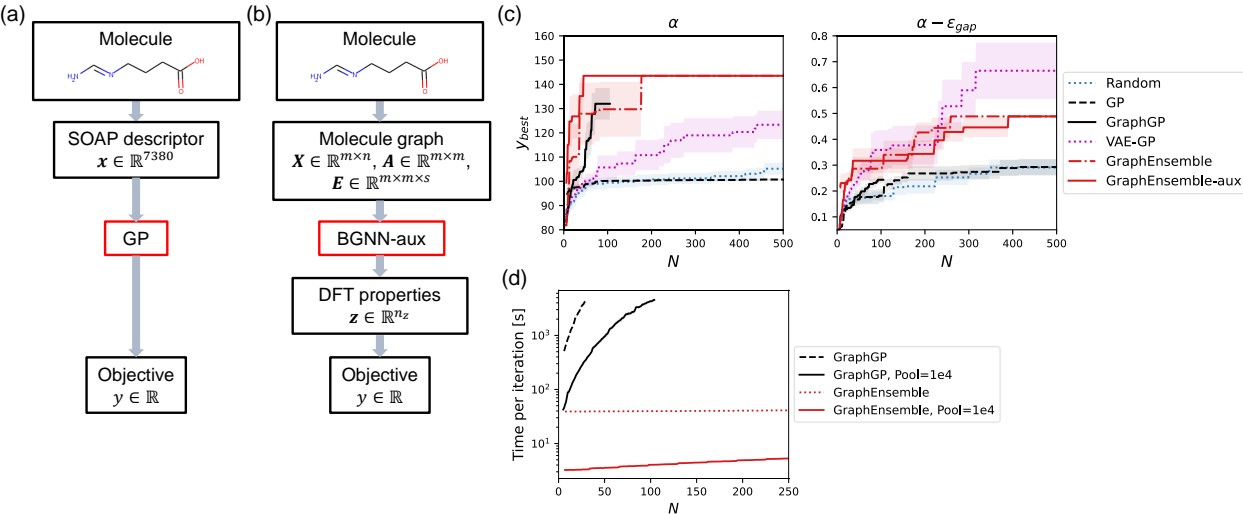

Figure 5: Quantum chemistry task and results. (a) The GP is trained on the SOAP descriptor, which is precomputed for each molecule. (b) The BGNN operates directly on a graph representation of the molecule, where atoms and bonds are represented by nodes and edges, respectively. The BGNN can be trained on multiple properties given in the QM9 dataset. (c) BO results for various properties. Note that GraphEnsemble is a type of BGNN. (d) Time per BO iteration on the GPU. (Note the logarithmic scale on the y-axis.) GRAPHGP takes orders of magnitudes longer than BGNNs for moderate $N$.

networks), we use the Smooth Overlap of Atomic Positions (SOAP) descriptor to produce a fixed-length feature vector for each molecule, as shown in Figure 5(a) (De et al., 2016; Himanen et al., 2020).

We compare two different optimization objectives derived from the QM9 dataset: the isotropic polarizability $\alpha$ and $(\alpha - \epsilon_{\text{gap}})$ where $\epsilon_{\text{gap}}$ is the HOMO-LUMO energy gap. Other objectives are included in Appendix A.5.4. Because many of the chemical properties in the QM9 dataset can be collectively computed by a single DFT or molecular dynamics calculation, we can treat a group of labels from QM9 as auxiliary information $\mathbf{z}$ and train our BGNN to predict this entire group simultaneously. The objective function $h$ then simply picks out the property of interest.

As shown in Figure 5(c), GRAPHGP and the BGNN variants significantly outperform GPs, showing that the inductive bias in the graph structure leads to a much more natural representation of the molecule and its properties. In the case of maximizing the polarizability $\alpha$, including the auxiliary information improves BO performance, showing signs of positive transfer. However, it does not have a significant impact on the other objectives, which may be due to the small size of the available auxiliary information (only a handful of chemical properties from the QM dataset) as compared with the nanoparticle and photonic crystal tasks. In a more realistic online setting, we would have significantly more physically-informative information available from a DFT calculation, e.g. we could easily compute the electronic density of states (the electronic analogue of the auxiliary information used in the photonics task).

As seen in Figure 5(d), we also note that the GRAPHGP is relatively computationally expensive ($15\times$ longer than GPs for small $N$ and $800\times$ longer than BGNNs for $N = 100$) and so we are only able to run it for a limited $N$ in a reasonable time frame. We see that BGNNs perform comparably or better than GRAPHGPs despite incurring a fraction of the computational cost.

VAE-GP uses a modified version of the latent-space optimization method implementation provided by Tripp et al. (2020). Rather than optimizing over a continuous latent space of the VAE, we feed the data pool through the VAE encoder to find their latent space representation, and then apply the acquisition function to the latent points to pick out the best unlabeled point to sample. We keep as many hyper-parameters the same as the original implementation as possible, with the exception of the weighted re-

training which we forgo since we have a fixed data pool that was used to train the VAE. This setup is similar to GraphNeuralLinear in that a deep learning architecture is used to encode the molecule as a continuous vector, although GraphNeuralLinear is only trained on the labelled data. The results for this experiment show that VAE-GP performs worse than BNNs on two of the three objective functions we tested and slightly better on one objective. We also note that the performance of VAE-GP depends very heavily on the pre-training of the VAE, as choosing different hyper-parameters or even a different random seed can significantly deteriorate performance (see Figure 15 in the Appendix).

## 5 Discussion

Introducing physics-informed priors (in the form of inductive biases) into the model are critical for their performance. Well-known inductive biases in deep learning include convolutional and graph neural networks for images and graph structures, respectively, which significantly improve BO performance. Another inductive bias that we introduce is the addition of auxiliary information present in composite functions, which significantly improves the performance of BO for the nanoparticle and photonic crystal tasks. We conjecture that the additional information forces the BNN to learn a more consistent physical model of the system since it must learn features that are shared across the multi-dimensional auxiliary information, thus enabling the BNN to generalize better. For example, the scattering spectrum of the multilayer particle consists of multiple resonances (sharp peaks), the width and location of which are determined by the material properties and layer thicknesses. The BNN could potentially learn these more abstract features, and thus, the deeper physics, to help it interpolate more efficiently, akin to data augmentation (Peurifoy et al., 2018). Auxiliary information can also be interpreted as a form of data augmentation. Indeed, tracking the prediction error on a validation set shows that models with auxiliary information tend to have a lower loss than those without (see Appendix A.5). It is also possible that the loss landscape for the auxiliary information is smoother than that of the objective function and that the auxiliary information acts as an implicit regularization that improves generalization performance.

Interestingly, GP-aux performs extremely poorly on the nanoparticle and photonic crystal tasks. One possible reason is that we are only able to run GP-aux for a few iterations, and it is not uncommon for GP-based BO to require some critical number of iterations to reach convergence especially in the case of high-dimensional systems where the size of the covariance matrix scales with the square of the dimensionality. It may also be possible that GP-aux only works on certain types of decompositions of functions and cannot be applied broadly to all composite functions, as the inductive biases in GPs are often hard-coded.

There is an interesting connection between how well BNNs are able to capture and explore a multi-modal posterior distribution and their performance in BO. For example, we have noticed that larger batch sizes tend to significantly hurt BO performance. On the one hand, larger batch sizes may be resulting in poorer generalization as the model finds sharper local minima in the loss landscape. Another explanation is that the stochasticity inherent in smaller batch sizes allows the BNN to more easily explore the posterior distribution, which is known to be highly multi-modal (Fort et al., 2019). Indeed, BO often underperforms for very small dataset sizes $N$ but quickly catches up as $N$ increases, indicating that batch size is an important hyperparameter which must be balanced with computational cost.

All our results use continued training (or warm restart) to minimize training costs. We note that re-initializing $\mathcal{M}$ and training from scratch in every iteration performs better than continued training on some tasks (results in the Appendix), which points to how BNNs may not sufficiently represent a multi-modal posterior distribution or that continued training may skew the training distribution that the BNN sees. Future work will consider using stochastic training approaches such as SG-MCMC methods for exploring posterior distributions (Welling & Teh, 2011; Zhang et al., 2019b) as well as other continual learning techniques to further minimize training costs, especially for larger datasets (Parisi et al., 2019).

When comparing BNN architectures, we find that ensembles tend to consistently perform among the best, which is supported by previous literature showing that ensembles capture uncertainty much better than variational methods (Ovadia et al., 2019; Gustafsson et al., 2020) especially in multi-modal loss landscapes (Fort et al., 2019). Ensembles are also attractive because they require no additional hyperparameters and they are simple to implement. Although training costs increase linearly with the size of the ensemble, this

can be easily parallelized on modern computing infrastructures. Furthermore, recent work that aims to model efficient ensembles that minimize computational cost could be an interesting future direction (Havasi et al., 2020; Wen et al., 2020). NEURALLINEAR variants are also quite powerful and cheap, making them very promising for tasks without high-dimensional auxiliary information. Integrating Neural Linear with multi-output GPs is an interesting direction for future work. The other BNNs either require extensive hyper-parameter tuning or perform poorly, making them difficult to use in practice. Additional discussion can be found in Appendix A.5.5.

As seen in Appendix A.5.4, VAE-GP performs worse than our method on two of the chemistry objectives and better on one objective. While latent-space optimization methods are often applied to domains where one wants to simultaneously generate data and optimize over the data distribution, these methods can also be applied to the cases in this work, where a data pool (e.g. QM9 dataset for the chemistry task) or separate data generation process (e.g. level-set process for the photonic crystal task) is already available. In these cases, the VAE is not used as a generative model, but rather as a way to learn appropriate representations. While latent-space approaches are able to take advantage of well-developed and widely available optimization algorithms, they also require unsupervised pre-training on a sizeable dataset and a suitable autoencoder model with the necessary inductive biases. Such models are available in chemistry where there have been significant development, but are more limited in other domains such as photonics. On the other hand, our method is able to incorporate the data structure or domain knowledge in an end-to-end manner during training, although future work is needed to more carefully evaluate how much of an advantage this is and whether it depends on specific dataset or domain characteristics. For settings where we do not need a generative model, it would also be interesting to replace the autoencoder with a self-supervised model (Hendrycks et al., 2019; Loh et al., 2021) or semi-supervised model (Kingma et al., 2014) to create a suitable latent space.

## 6 Conclusion

We have demonstrated global optimization on multiple tasks using a combination of deep learning and BO. In particular, we have shown how BNNs can be used as surrogate models in BO, which enables the scaling of BO to large datasets and provides the flexibility to incorporate a wide variety of constraints, data augmentation techniques, and inductive biases. We have demonstrated that integrating domain-knowledge on the structure and symmetries of the data into the surrogate model as well as exploiting intermediate or auxiliary information significantly improves BO performance, all of which can be interpreted as physics-informed priors. Intuitively, providing the BNN surrogate model with all available information allows the BNN to learn a more faithful physical model of the system of interest, thus enhancing the performance of BO. Finally, we have applied BO to real-world, high-dimensional scientific datasets, and our results show that BNNs can outperform our best-effort GPs, even with strong domain-dependent structure encoded in the covariance functions. We note that our method is not necessarily tied to any particular application domain, and can lower the barrier of entry for design and optimization.

Future work will investigate more complex BNN architectures with stronger inductive biases. For example, output constraints can be placed through unsupervised learning (Karpatne et al., 2017) or by variationally fitting a BNN prior (Yang et al., 2020). Custom architectures have also been proposed for partial differential equations (Raissi et al., 2017; Lu et al., 2020), many-body systems (Cranmer et al., 2020), and generalized symmetries (Hutchinson et al., 2020), which will enable effective BO on a wider range of tasks. The methods and experiments presented here enable BO to be effectively applied in a wider variety of settings. There are also variants of BO including TURBO which perform extremely well on our tasks, and so future work will also include incorporating BNNs into these variants.

We make our datasets and code publicly available at `https://github.com/samuelkim314/DeepBO`

**Acknowledgements**

The authors would like to acknowledge Rodolphe Jenatton, Thomas Christensen, Andrew Ma, Rumen Dangovski, Joy Zeng, Tin D. Nguyen, Charles Roques-Carmes, and Mohammed Benzaouia for fruitful conversations. The authors acknowledge the MIT SuperCloud and Lincoln Laboratory Supercomputing Center for

providing HPC resources that have contributed to the research results reported within this paper. This work is supported in part by the the National Science Foundation under Cooperative Agreement PHY-2019786 (The NSF AI Institute for Artificial Intelligence and Fundamental Interactions, `http://iaifi.org/`). This research was also sponsored in part by the Department of Defense through the National Defense Science & Engineering Graduate Fellowship (NDSEG) Program. This material is based upon work partly supported by the Air Force Office of Scientific Research under the award number FA9550-21-1-0317, as well partly supported by the US Office of Naval Research (ONR) Multidisciplinary University Research Initiative (MURI) grant N0 0014-20-1-2325 on Robust Photonic Materials with High-Order Topological Protection It is also based upon work supported in part by the U.S. Army Research Office through the Institute for Soldier Nanotechnologies at MIT, under Collaborative Agreement Number W911NF-18-2-0048. Research was sponsored by the United States Air Force Research Laboratory and the United States Air Force Artificial Intelligence Accelerator and was accomplished under Cooperative Agreement Number FA8750-19-2-1000. The views and conclusions contained in this document are those of the authors and should not be interpreted as representing the official policies, either expressed or implied, of the United States Air Force or the U.S. Government. The U.S. Government is authorized to reproduce and distribute reprints for Government purposes notwithstanding any copyright notation herein.

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

Table 1: Summary of dataset dimensionalities. Note that alternate inputs for photonic crystal and organic molecule datasets are binary images and molecule graphs, respectively.

|  | CONTINUOUS INPUT DIMENSION | ALTERNATE INPUT DIMENSION | AUXILIARY DIMENSION |
|---|---|---|---|
| NANOPARTICLE SCATTERING | 6 | N/A | 201 |
| PHOTONIC CRYSTAL DOS | 51 | $32 \times 32 = 1024$ | 500 |
| MOLECULE QUANTUM CHEMISTRY | 480 | $9 + 9 \times 9 + 9 \times 9 = 171$ | 9 |

# A Appendix

## A.1 Datasets

The dimensionalities of the datasets are summarized in table 1. The continuous input dimension for chemical molecules refers to the SOAP descriptor. While the space of chemical molecule graphs in general do not have a well-defined dimensionality as chemical molecules can be arbitrarily large and complex, we limit the size of molecules by only sampling from the QM9 dataset, and can define the dimensionality as the sum of the adjacency, node, and edge matrix dimensionalities.

The high dimensionalities of all of these problems make Bayesian neural networks well-suited as surrogate models to enable scaling. Note that the nanoparticle scattering problem can be adjusted to be less or more difficult by either changing the input dimensionality (i.e. the number of nanoparticle layers) or the auxiliary dimension (i.e. the resolution or range of wavelengths that are sampled).

### A.1.1 Nanoparticle Scattering

The multilayer nanoparticle consists of a lossless silica core surrounded by alternating spherical layers of lossless $TiO_2$ and lossless silica. The relative permittivity of silica is $\varepsilon_{\text{silica}} = 2.04$. The relative permittivity of $TiO_2$ is dispersive and depends on the wavelength of light:

$$\varepsilon_{\text{TiO}_2} = 5.913 + \frac{0.2441}{10^{-6}\lambda^2 - 0.0803} \tag{6}$$

where $\lambda$ is the wavelength given in units of nm. The entire particle is surrounded by water, which has a relative permittivity of $\varepsilon_{\text{water}} = 1.77$.

For a given set of thicknesses, we analytically solve for the scattering spectrum, i.e. the scattering cross-section $\sigma(\lambda)$ as a function of wavelength $\lambda$, using Mie scattering as described in Qiu et al. (2012). The code for computing $\sigma$ was adapted from Peurifoy et al. (2018).

The objective functions for the narrowband and highpass objectives are:

$$h_{\text{nb}}(\mathbf{z}) = \frac{\int_{\lambda \in \text{nb}} \sigma(\lambda)\,d\lambda}{\int_{\text{elsewhere}} \sigma(\lambda)\,d\lambda} \approx \frac{\sum_{i=126}^{145} z_i}{\sum_{i=1}^{125} z_i + \sum_{i=146}^{201} z_i} \tag{7}$$

$$h_{\text{hp}}(\mathbf{z}) = \frac{\int_{\lambda \in \text{hp}} \sigma(\lambda)\,d\lambda}{\int_{\text{elsewhere}} \sigma(\lambda)\,d\lambda} \approx \frac{\sum_{i=126}^{201} z_i}{\sum_{i=1}^{125} z_i} \tag{8}$$

where $\mathbf{z} \in \mathbb{R}^{201}$ is the discretized scattering cross-section $\sigma(\lambda)$ from $\lambda = 350\,\text{nm}$ to $750\,\text{nm}$.

### A.1.2 Photonic Crystal

The photonic crystal (PC) consists of periodic unit cells with periodicity $a = 1\,\text{au}$, where each unit cell is depicted as a "two-tone" image, with the white regions representing silicon with permittivity $\varepsilon_1 = 11.4$ and black regions representing vacuum (or air) with permittivity $\varepsilon_0 = 1$.

The photonic crystal (PC) structure is defined by a spatially varying permittivity $\varepsilon(x, y) \in \{\varepsilon_0, \varepsilon_1\}$ over a 2D periodic unit cell with spatial coordinates $x, y \in [0, a]$. To parameterize $\varepsilon$, we choose a level set of a

Fourier sum function $\phi$, defined as a linear combination of plane waves with frequencies evenly spaced in the reciprocal lattice space up to a maximum cutoff. Intuitively, the upper limit on the frequencies roughly corresponds to a lower limit on the feature size such that the photonic crystal remains within reasonable fabrication constraints. Here we set the cutoff such that there are 25 complex frequencies corresponding to 50 real coefficients $\mathbf{c} = (c_1, c_2, ..., c_{50})$.

Explicitly, we have

$$\phi[\mathbf{c}](x, y) = \Re \left\{ \sum_{k=1}^{25} (c_k + ic_{k+25}) \, e^{2\pi i (n_x x + n_y y)/a} \right\}, \tag{9}$$

where each exponential term is composed from the 25 different pairs $\{n_x, n_y\}$ with $n_x, n_y \in \{-2, -1, 0, 1, 2\}$. We then choose a level-set offset $\Delta$ to determine the PC structure, where regions with $\phi > \Delta$ are assigned to be silicon and regions where $\phi \leq \Delta$ are vacuum. Thus, the photonic crystal unit cell topology is parameterized by a 51-dimensional vector, $[c_1, c_2, ..., c_{50}, \Delta] \in \mathbb{R}^{51}$. More specifically,

$$\varepsilon(x, y) = \varepsilon[\mathbf{c}, \Delta](x, y) = \begin{cases} \varepsilon_1 & \phi[\mathbf{c}](x, y) > \Delta \\ \varepsilon_0 & \phi[\mathbf{c}](x, y) \leq \Delta \end{cases}, \tag{10}$$

which is discretized to result in a $32 \times 32$ pixel image $\mathbf{v} \in \{\varepsilon_0, \varepsilon_1\}^{32 \times 32}$. This formulation also has the advantage of enforcing periodic boundary conditions.

For each unit cell, we use the MIT Photonics Bands (MPB) software (Johnson & Joannopoulos, 2001) to compute the band structure of the photonic crystal, $\omega(\mathbf{k})$, up to the lowest 10 bands, using a $32 \times 32$ spatial resolution (or equivalently, $32 \times 32$ k-points over the Brillouin zone $-\frac{\pi}{a} < k < \frac{\pi}{a}$). We also extract the group velocities at each k-point and compute the density-of-states (DOS) via an extrapolative technique, adapted from Liu et al. (2018). The DOS is computed at a resolution of 20,000 points, and a Gaussian filter of kernel size 100 is used to smooth the DOS spectrum. To normalize the frequency scale across the different unit cells, the frequency is rescaled via $\omega \sqrt{\varepsilon_{avg}} \to \omega_{norm}$, where $\varepsilon_{avg} = \frac{1}{a^2} \int_0^a \int_0^a \varepsilon(x, y) \, dx \, dy \approx \frac{1}{(32)^2} \sum_{i,j} v_{i,j}$ is the average permittivity over all pixels. Finally, the DOS spectrum is truncated at $\omega_{norm} = 1.2$ and interpolated using 500 points to give $\mathbf{z} \in \mathbb{R}^{500}$.

The objective function aims to minimize the DOS in a small frequency range and maximize it elsewhere. We use the following:

$$h_{\text{DOS}}(\mathbf{z}) = \frac{\sum_{i=1}^{300} z_i + \sum_{i=351}^{500} z_i}{1 + \sum_{i=301}^{350} z_i}, \tag{11}$$

where the 1 is added in the denominator to avoid singular values.

### A.1.3 Organic Molecule Quantum Chemistry

The Smooth Overlap of Atomic Positions (SOAP) descriptor (De et al., 2016) uses smoothed atomic densities to describe local environments for each atom in the molecule through a fixed-length feature vector, which can then be averaged over all the atoms in the molecule to produce a fixed-length feature vector for the molecule. This descriptor is invariant to translations, rotations, and permutations. We use the SOAP descriptor implemented by DScribe (Himanen et al., 2020) using the parameters: local cutoff `rcut = 5`, number of radial basis functions `nmax = 3`, and maximum degree of spherical harmonics `lmax = 3`. We use `outer` averaging, which averages over the power spectrum of different sites.

The graph representation of each molecule is processed by the Spektral package (Grattarola & Alippi, 2020). Each graph is represented by a node feature matrix $\mathbf{X} \in \mathbb{R}^{s \times d_n}$, an adjacency matrix $\mathbf{A} \in \mathbb{R}^{s \times s}$, and an edge matrix $\mathbf{E} \in \mathbb{R}^{e \times d_e}$, where $s$ is the number of atoms in the molecule, $e$ is the number of bonds, and $d_n, d_e$ are the number of features for nodes and edges, respectively.

The properties that we use from the QM9 dataset are listed in Table 2. We separate these properties into two categories: (1) the *ground state quantities* which are calculated from a single DFT calculation of the molecule and include geometric, energetic, and electronic quantities, and (2) the *thermodynamic quantities* which are typically calculated from a molecular dynamics simulation.

Table 2: List of properties from the QM9 dataset used as labels

| PROPERTY | UNIT | DESCRIPTION |
|---|---|---|
| *GROUND STATE QUANTITIES* | | |
| $A$ | GHz | ROTATIONAL CONSTANT |
| $B$ | GHz | ROTATIONAL CONSTANT |
| $C$ | GHz | ROTATIONAL CONSTANT |
| $\mu$ | D | DIPOLE MOMENT |
| $\alpha$ | $a_0^3$ | ISOTROPIC POLARIZABILITY |
| $\epsilon_{\text{HOMO}}$ | Ha | ENERGY OF HOMO |
| $\epsilon_{\text{LUMO}}$ | Ha | ENERGY OF LUMO |
| $\epsilon_{\text{GAP}}$ | Ha | GAP ($\epsilon_{\text{LUMO}} - \epsilon_{\text{HOMO}}$) |
| $\langle R^2 \rangle$ | $a_0^2$ | ELECTRONIC SPATIAL EXTENT |
| *THERMODYNAMIC QUANTITIES AT* 298.15 K | | |
| $U$ | Ha | INTERNAL ENERGY |
| $H$ | Ha | ENTHALPY |
| $G$ | Ha | FREE ENERGY |
| $C_V$ | $\frac{\text{cal}}{\text{mol K}}$ | HEAT CAPACITY |

---

**Algorithm 1** Bayesian optimization with auxiliary information

---

1: **Input:** Labelled dataset $\mathcal{D}_{\text{train}} = \{(\mathbf{x}_n, \mathbf{z}_n, y_n)\}_{n=1}^{N_{\text{start}}=5}$
2: **for** $N = 5$ **to** 1000 **do**
3:     Train $\mathcal{M}: \mathcal{X} \to \mathcal{Z}$ on $\mathcal{D}_{\text{train}}$
4:     Form an unlabelled dataset, $\mathcal{X}_{\text{pool}}$
5:     Find $\mathbf{x}_{N+1} = \arg\max_{\mathbf{x} \in \mathcal{X}_{\text{pool}}} \alpha\left(\mathbf{x}; \mathcal{M}, \mathcal{D}_{\text{train}}\right)$
6:     Label the data $\mathbf{z}_{N+1} = g(\mathbf{x}_{N+1})$, $y_{N+1} = h(\mathbf{z}_{N+1})$
7:     $\mathcal{D}_{\text{train}} = \mathcal{D}_{\text{train}} \cup (\mathbf{x}_{N+1}, \mathbf{z}_{N+1}, y_{N+1})$
8: **end for**

---

The auxiliary information for this task consist of the properties listed in Table 2 that are in the same category as the objective property, as these properties would be calculated together. The objective function then simply picks out the corresponding feature from the auxiliary information. More precisely, for the ground state objectives, the auxiliary information is

$$\mathbf{z} = \left[A, B, C, \mu, \alpha, \epsilon_{\text{HOMO}}, \epsilon_{\text{LUMO}}, \epsilon_{\text{gap}}, \langle R^2 \rangle\right] \in \mathbb{R}^9,$$

and the objective functions are

$$h_\alpha(\mathbf{z}) = z_5$$
$$h_{\alpha-\epsilon_{\text{gap}}}(\mathbf{z}) = \frac{z_5 - 6}{191} - \frac{z_8 - 0.02}{0.6}$$

where the quantities for the latter objective are normalized so that they have the same magnitude.

### A.2 Bayesian Optimization and Acquisition Function

Our algorithm for Bayesian optimization using auxiliary information $\mathbf{z}$ is shown in Algorithm 1. This algorithm reduces to the basic BO algorithm in the case where $h$ is the identity function and $\mathcal{Z} = \mathcal{Y}$ such that we can ignore mention of $\mathbf{z}$ in Algorithm 1.

As mentioned in the main text, the inner optimization loop in line 5 of Algorithm 1 is performed by finding the maximum value of $\alpha$ over a pool of $|\mathcal{X}_{\text{pool}}|$ randomly sampled points. We can see in Figure 6 that increasing $|\mathcal{X}_{\text{pool}}|$ in the acquisition step tends to improve BO performance. Thus, there is likely further room for improvement of the inner optimization loop using more sophisticated algorithms, possibly using the gradient information provided by BNNs. Unless otherwise stated, we optimize the inner loop of Bayesian

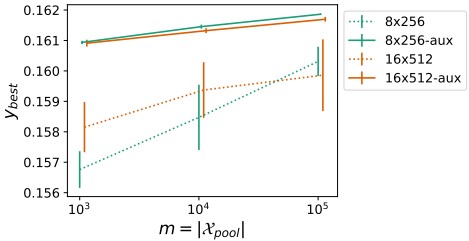

Figure 6: Effect of $m = |\mathcal{X}_{\text{pool}}|$ used in the inner optimization loop to maximize the acquisition function on overall BO performance. $y_{\text{best}}$ is taken from the narrowband objective function using the ensemble architecture. The "aux" in the legend denotes using auxiliary information and the numbers represent the architecture (i.e. 8 layers of 256 units or 16 layers of 512 units).

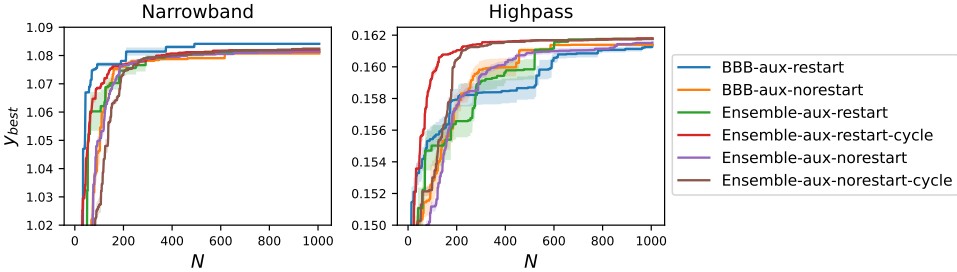

Figure 7: Effect of restarting the BNN training from scratch in each BO iteration.

optimization to choose the next data point to label by maximizing EI on a pool of $|\mathcal{X}_{\text{pool}}| = 10^5$ randomly sampled points.

### A.3 Continued Training

As mentioned in Section 2.3 of the main text, the BNN is ideally trained from scratch until convergence in each iteration loop, although this comes at a great computational cost. An alternative is the warm restart method of continuing the training from the previous iteration which enables the model's training loss to converge in only a few epochs. However, as shown in Figure 7, we have found that naive continued training can result in poor BO performance. This is likely because (a) training does not converge for the new data point $\mathcal{D}_{\text{new}} = (\mathbf{x}_{N+1}, y_{N+1})$ relative to the rest of the data under a limited computational budget, resulting in the acquisition function possibly labeling similar points in consecutive iterations, and (b) the BNN gets trapped in a local minima in the loss landscape that is not ideal for learning future data points. To mitigate this, we use the cosine annealing learning rate proposed in Loshchilov & Hutter (2016). The large learning rate at the start of training allows the model to more easily escape local minima and explore a multimodal posterior (Huang et al., 2017), while the small learning rate towards the end of the annealing cycle allows the model to converge more easily. Note that the idea of warm restart is similar to "continual learning," which is an open and active sub-problem in machine learning research (Thrun, 1998; Parisi et al., 2019). In particular, we re-train the BNN using 10 epochs.

### A.4 Models and Hyperparameters

### A.4.1 Additional Surrogate Models

**Variational BNNs** model a prior and posterior distribution over the neural network weights, but use some approximation on the distributions to make the BNN tractable. In particular, we use Bayes by Backprop

(BBB) (also referred to as the "mean field" approximation), which approximates the posterior over the neural network weights with independent normal distributions (Blundell et al., 2015). We also compare Multiplicative Normalizing Flows (MNF), which uses normalizing flows on top of each layer output for more expressive posterior distributions (Louizos & Welling, 2017).

**BOHAMIANN** proposed to use BNNs in BO by using stochastic gradient Hamiltonian Monte Carlo (SGHMC) to approximately sample the BNN posterior, combined with scale adaptation to adapt it for an iterative setting (Springenberg et al., 2016).

**NeuralLinear** trains a conventional neural network on the data, but then replaces the last layer with Bayesian linear regression such that the neural network serves as an adaptive basis for the linear regression (Snoek et al., 2015).

**TuRBO** (trust region Bayesian Optimization) is a method that maintains $M$ trust regions and performs Bayesian optimization within each trust region, maintaining $M$ local surrogate models, to scale BO to high-dimensional problems that require thousands of observations (Eriksson et al., 2019). We use $M = 1$ and $M = 5$, labelled as "TuRBO-1" and "TuRBO-5", respectively.

**TPE** (Tree Parzen Estimator) is a method that instead of modeling $p(y|x)$, models $p(x|y)$ and $p(y)$ for the surrogate model and fits into the BO framework (Bergstra et al., 2013). The tree-structure of the surrogate model allows it to define leaf variables only when node variables take particular values, which makes it well-suited for hyper-parameter search (e.g. the learning rate momentum is only defined for momentum-based gradient descent methods).

**LIPO** is a parameter-free algorithm that assumes the underlying function is a Lipschitz function and estimates the bounds of the function (Malherbe & Vayatis, 2017). We use the implementation provided by the dlib library (King, 2009).

**DIRECT-L** (DIviding RECTangles-Local) systematically divides the search domain into smaller and smaller hyperrectangles to efficiently search the space (Gablonsky & Kelley, 2001). We use the implementation provided by the NLopt library (Johnson, 2010).

**CMA-ES** (covariance matrix adaptation evolution strategy) is an evolutionary algorithm that samples new data based on a multivariate normal distribution and refines the parameters of this distribution until reaching convergence. We us the implementation provided by the pycma library (Hansen et al., 2019).

### A.4.2 Implementation Details

Unless otherwise stated, we set $N_{\mathrm{MC}} = 30$. All BNNs other than the infinitely-wide networks are implemented in TensorFlow v1. Models are trained using the Adam optimizer using the cosine annealing learning rate with a base learning rate of $10^{-3}$ (Loshchilov & Hutter, 2016). All hidden layers use ReLU as the activation function, and no activation function is applied to the output layer.

Infinite-width neural networks are implemented using the Neural Tangents library (Novak et al., 2020). We use two different types of infinite networks: (1) "GP-" refers to a closed form expression for Gaussian process inference using the infinite-width neural network as a kernel, and (2) "Inf-" refers to an infinite ensemble of infinite-width networks that have been "trained" with continuous gradient descent for an infinite time. We compare NNGP and NTK kernels as well as the parameterization of the layers. By default, we use the NTK parameterization, but we also use the standard parameterization, denoted by "-std".

We implement BO using GPs with a Matérn kernel using the GPyOpt library (The GPyOpt authors, 2016). The library optimizes over the acquisition function in the inner loop using the L-BFGS algorithm.

LIPO (Malherbe & Vayatis, 2017) is implemented in the dlib library (King, 2009). DIRECT-L (Gablonsky & Kelley, 2001) is implemented in the NLopt library (Johnson, 2010). CMA-ES is implemented in the pycma library (Hansen et al., 2019).

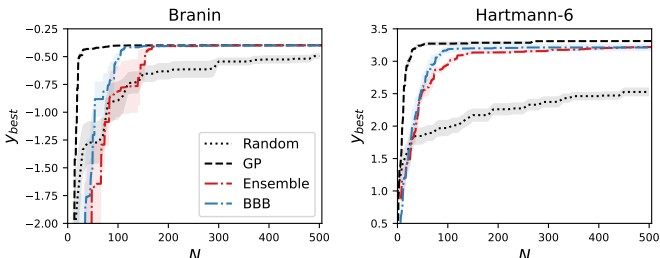

Figure 8: BO results for the Branin and Hartmann-6 functions.

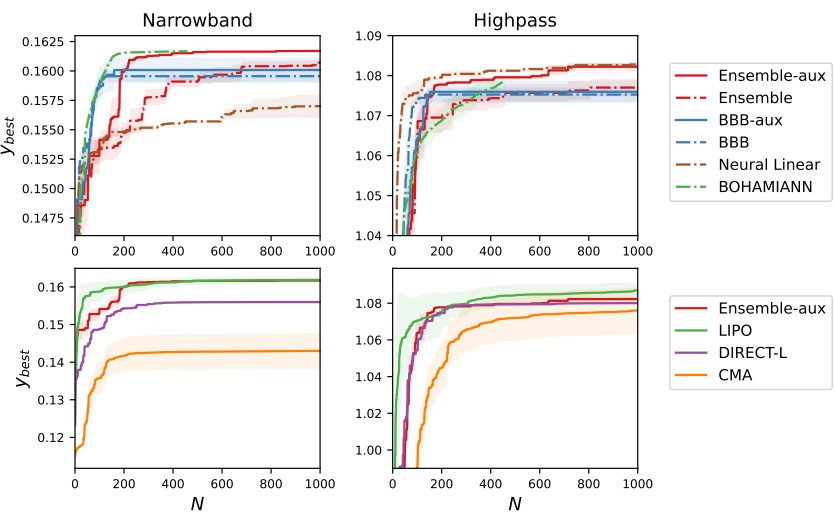

Figure 9: Additional optimization result curves for the nanoparticle scattering task. (Top) Various BNNs. Note that results using auxiliary information are denoted by a solid line, while those that do not are denoted by a dashed line. Also note that the y-axis is zoomed in to differentiate the curves. (Bottom) Various non-BO algorithms. ENSEMBLE-AUX is replicated here for ease of comparison.

## A.5 Additional Results

### A.5.1 Test Functions

We test BO on several common synthetic functions used for optimization, namely the Branin and 6-dimensional Hartmann functions. We use BNNs with 4 hidden layers and 256 units in each hidden layer, where each hidden layer is followed by a ReLU activation function. Plots of the best value $y_{\text{best}}$ at each BO iteration are shown in Figure 8. As expected, GPs perform the best. Ensembles and BBB also perform competitively and much better than random sampling, showing that deep BO is viable even for low-dimensional black-box functions.

### A.5.2 Nanoparticle Scattering

Detailed BO results for the nanoparticle scattering problem are shown in Table 3.

All the BNNs used for the nanoparticle scattering problem use an architecture consisting of 8 hidden layers with 256 units each, with the exception of BOHAMIANN where we used the original architecture consisting of 2 hidden layers with 50 units each. The infinite-width neural networks for the nanoparticle task consist of 8 hidden layers of infinite width, each of which are followed by ReLU activation functions.

Table 3: BO results for the nanoparticle scattering problem. * denotes that $y_{\text{best}}$ is measured at $N = 100$ due to computational constraints

| MODEL | NARROWBAND | | | | HIGHPASS | | | |
|---|---|---|---|---|---|---|---|---|
| | $y_{\text{BEST}}$ AT $N = 250$ | | $y_{\text{BEST}}$ AT $N = 1000$ | | $y_{\text{BEST}}$ AT $N = 250$ | | $y_{\text{BEST}}$ AT $N = 100$ | |
| | MEAN | SE | MEAN | SE | MEAN | SE | MEAN | SE |
| GP | 0.1606 | 0.0005 | 0.1621 | 0.0001 | 1.0839 | 0.0017 | 1.0851 | 0.0008 |
| GP-AUX | *0.1541 | 0.0019 | - | - | *1.0110 | 0.0234 | - | - |
| ENSEMBLE | 0.1558 | 0.0011 | 0.1607 | 0.0003 | 1.0729 | 0.0025 | 1.077 | 0.0021 |
| ENSEMBLE-AUX | 0.1578 | 0.0014 | 0.1593 | 0.0013 | 1.0783 | 0.0003 | 1.0822 | 0.001 |
| BBB | 0.1596 | 0.0006 | 0.1596 | 0.0006 | 1.0753 | 0.0005 | 1.0753 | 0.0005 |
| BBB-AUX | 0.1601 | 0.001 | 0.1601 | 0.001 | 1.076 | 0.0028 | 1.076 | 0.0028 |
| BBB-ANNEAL | 0.1598 | 0.001 | 0.1611 | 0.0001 | 1.0813 | 0.0003 | 1.0821 | 0.0005 |
| BBB-AUX-ANNEAL | 0.1613 | 0.0003 | 0.1619 | 0 | 1.0826 | 0.0008 | 1.0834 | 0.0005 |
| MNF | 0.15 | 0.0005 | 0.1547 | 0.0004 | 1.027 | 0.005 | 1.0312 | 0.0036 |
| MNF-AUX | 0.1549 | 0.0014 | 0.1569 | 0.0006 | 0.9957 | 0.0168 | 1.028 | 0.0157 |
| NEURAL LINEAR | 0.1543 | 0.002 | 0.1579 | 0.0015 | 1.0798 | 0.0007 | 1.0836 | 0.0007 |
| BOHAMIANN | 0.1616 | 0.0001 | - | - | 1.0717 | 0.0031 | - | - |
| INF-NNGP | 0.1541 | 0.0011 | 0.157 | 0.0009 | 1.055 | 0.0036 | 1.0653 | 0.0022 |
| INF-NTK | 0.1536 | 0.0008 | 0.1571 | 0.001 | 1.041 | 0.004 | 1.0612 | 0.0011 |
| INF-NNGP-STD | 0.1551 | 0.0006 | 0.1598 | 0.0006 | 1.0615 | 0.0043 | 1.069 | 0.0018 |
| INF-NTK-STD | 0.1564 | 0.0006 | 0.1607 | 0.0001 | 1.0607 | 0.0039 | 1.0761 | 0.0014 |
| GP-NNGP | 0.1582 | 0.0007 | 0.1609 | 0.0001 | 1.0621 | 0.0027 | 1.0694 | 0.0019 |
| GP-NTK | 0.1573 | 0.001 | 0.1611 | 0.0001 | 1.0667 | 0.0032 | 1.0732 | 0.0012 |
| GP-NNGP-STD | 0.1562 | 0.0008 | 0.1595 | 0.001 | 1.0615 | 0.0058 | 1.0718 | 0.0024 |
| GP-NTK-STD | 0.1592 | 0.0011 | 0.1608 | 0.0002 | 1.0641 | 0.0033 | 1.0704 | 0.0017 |
| TURBO-1 | 0.1572 | 0.0017 | 0.1619 | 0.0011 | 1.0831 | 0.022 | 1.0871 | 0.0005 |
| TURBO-1 | 0.1605 | 0.0011 | 0.1619 | 0.0001 | 1.0867 | 0.0016 | 1.0890 | 0.0003 |
| TPE | 0.1561 | 0.0007 | 0.1615 | 0.0001 | 1.0517 | 0.0035 | 1.0794 | 0.0010 |
| RANDOM | 0.1527 | 0.0008 | 0.1555 | 0.0006 | 1.0053 | 0.0063 | 1.0362 | 0.0047 |
| LIPO | 0.1604 | 0.0016 | 0.1619 | 0.0006 | 1.0792 | 0.0066 | 1.087 | 0.0034 |
| DIRECT-L | 0.1544 | 0 | 0.156 | 0 | 1.0777 | 0 | 1.0801 | 0 |
| CMA | 0.1424 | 0.0046 | 0.143 | 0.0048 | 1.059 | 0.0117 | 1.076 | 0.0127 |

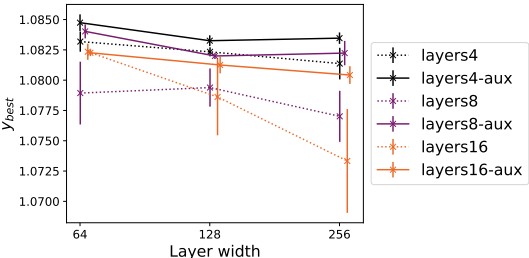

Figure 10: Comparison of $y_{\text{best}}$ at $N = 1000$ for the nanoparticle narrowband objective function for a variety of neural network sizes. All results are ensembles, and "aux" denotes using auxiliary information.

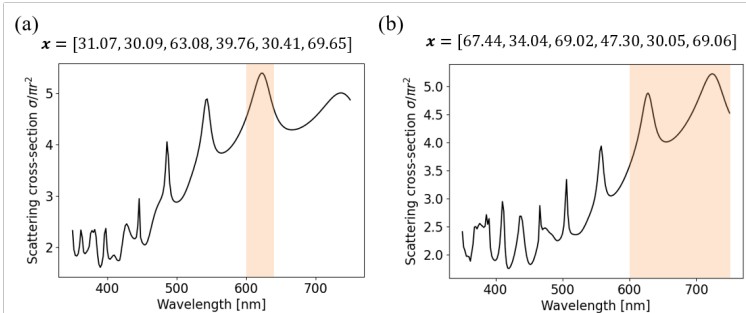

Figure 11: Examples of optimized nanoparticles and their scattering spectrum using the "Ensemble-aux" architecture for the (a) narrowband and (c) highpass objectives. Orange shaded regions mark the range over which we wish to maximize the scattering.

We also experiment with KL annealing in BBB, a proposed method to improve the performance of variational methods for BNNs in which the weight of the KL term in the loss function is slowly increased throughout training Wenzel et al. (2020). For these experiments, we exponentially anneal the KL term with weight $\sigma_{KL}(i) = 10^{i/500-5}$ as a function of epoch $i$ when training from scratch; during the continued training, the weight is held constant at $\sigma_{KL} = 10^{-3}$.

KL annealing in the BBB architecture significantly improves performance for the narrowband objective, although results are mixed for the highpass objective. Additionally, KL annealing has the downside of introducing more parameters that must be carefully tuned for optimal performance. MNF performs poorly, especially on the highpass objective where it is comparable to random sampling, and we have found that MNF is quite sensitive to the choice of hyperparameters for uncertainty estimates even on simple regression problems.

The different variants infinite-width neural networks do not perform as well as the BNNs on both objective functions, despite the hyper-parameter search.

LIPO seems to perform as well as GPs on both objective functions, which is impressive given the computational speed of the LIPO algorithm. Interestingly DIRECT-L does not perform as well as LIPO or GPs on the narrowband objective, and actually performs comparably to random sampling on the highpass objective. Additionally, CMA performs poorly on both objectives, likely due to the highly multimodal nature of the objective function landscape.

We also look at the effect of model size in terms of number of layers and units in Figure 10 for ensembles. While including auxiliary information clearly improves performance across all architectures, there is not a clear trend of performance with respect to the model size. Thus, the performance of BO seems to be somewhat robust to the exact architecture as long as the model is large enough to accurately and efficiently train on the data.

Table 4: Various architectures for BNNs and BCNNs used in the PC problem. Numbers represent the number of channels and units for the convolutional and fully-connected layers, respectively. All convolutional layers use $3 \times 3$-sized filters with stride $(1, 1)$ and periodic boundaries. "MP" denotes max-pooling layers of size $2 \times 2$ with stride $(2, 2)$, and "AP" denotes average-pooling layers of size $2 \times 2$ with stride $(1, 1)$. "CONV" denotes BCNNs whereas "FC" denotes BNNs (containing only fully-connected layers) that act on the level-set parameterization $\mathbf{x}$ rather than on the image $\mathbf{v}$. "TI" denotes translation invariant architectures, whereas "TD" denotes translation dependent architectures (i.e. not translation invariant).

| ARCHITECTURE | CONVOLUTIONAL LAYERS | FULLY-CONNECTED LAYERS |
|---|---|---|
| CONV-TI | 16-MP-32-MP-64-MP-128-MP-256 | 256-256-256-256 |
| CONV-TD | 8-AP-8-MP-16-AP-32-MP-32-AP | 256-256-256-256 |
| FC | *N/A* | 256-256-256-256-256 |

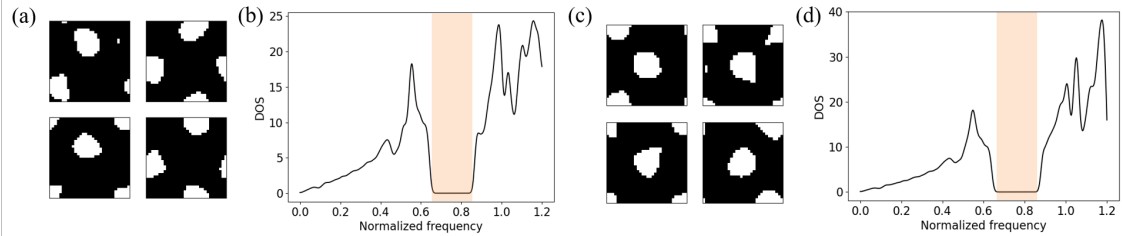

Figure 12: Examples of optimized photonic crystal unit cells over multiple trials for (a) PC-A distribution and (c) PC-B distribution. (b,d) Examples of the optimized DOS. Note that the DOS has been minimized to nearly zero in a thin frequency range. Orange shaded regions mark the frequency range in which we wish to minimize the DOS. All results were optimized by the "ENSEMBLE-AUX" architecture.

Examples of the optimized structures by the "ENSEMBLE-AUX" architecture are shown in Figure 11. We can see that the scattering spectra peak in the shaded region of interest, as desired by the respective objective functions.

### A.5.3 Photonic Crystal

The BNN and BCNN architectures that we use for the PC task are listed in Table 4. The size of the "FC" architectures are chosen to have a similar number of parameters as their convolutional counterparts. Unless otherwise stated, all results in the main text and here use the "CONV-TI" and "FC" architectures for BCNNs and BNNs, respectively.

The infinite-width convolutional neural networks (which act as convolutional kernels for GPs) in the PC task consist of 5 convolutional layers followed by 4 fully-connected layers of infinite width. Because the pooling layers in the Neural Tangents library are currently too slow for use in application, we increased the size of the filters to $5 \times 5$ to increase the receptive field of each filter.

Detailed BO results for the PC problem are shown in Table 5. For algorithms that optimize over the level set parameterization $\mathbb{R}^{51}$, we see that GPs perform consistently well, although BNNs using auxiliary information (e.g. Ensemble-Aux) can outperform GPs. DIRECT-L and CMA perform extremely well on the PC-A distribution but performs worse than GP on the PC-B distribution.

Adding convolutional layers and auxiliary information improves performance such that BCNNs significantly outperform GPs. Interestingly, the infinite-width networks perform extremely poorly, although this may be due to a lack of pooling layers in their architecture which limits the receptive field of the convolutions.

Examples of the optimized structures by the "ENSEMBLE-AUX" architecture are shown in Figure 12. The photonic crystal unit cells generally converged to the same shape: a square lattice of silicon posts with periodicity $\sqrt{2}a$.

Table 5: Select BO results for the PC problem. $^*$ denotes that $y_{\text{best}}$ is measured at $N = 130$ due to computational constraints. $^\dagger$ denotes that $y_{\text{best}}$ is measured at $N = 750$ due to computational constraints.

| Model | PC-A | | | | PC-B | | | |
|---|---|---|---|---|---|---|---|---|
| | $y_{\text{best}}$ at $N = 250$ | | $y_{\text{best}}$ at $N = 1000$ | | $y_{\text{best}}$ at $N = 250$ | | $y_{\text{best}}$ at $N = 100$ | |
| | Mean | SE | Mean | SE | Mean | SE | Mean | SE |
| GP | 548 | 450 | 2109 | 448 | 781 | 394 | 3502 | 49 |
| GP-aux | $^*$16 | 4 | - | - | $^*$9 | 1 | - | - |
| Ensemble | 30 | 2 | 841 | 448 | 216 | 145 | 1318 | 465 |
| Ensemble-aux | 305 | 217 | 1310 | 509 | 2909 | 408 | 3633 | 130 |
| ConvEnsemble | 1140 | 471 | 2375 | 371 | 390 | 263 | 2070 | 505 |
| ConvEnsemble-aux | 2623 | 558 | 3468 | 120 | 3752 | 106 | 4002 | 92 |
| BBB | 75 | 31 | 350 | 207 | 704 | 502 | 780 | 485 |
| BBB-aux | 39 | 7 | 413 | 313 | 554 | 371 | 1605 | 544 |
| ConvBBB | 712 | 416 | 1486 | 490 | 928 | 600 | 930 | 599 |
| ConvBBB-aux | 2109 | 583 | 3124 | 43 | 1761 | 724 | 1928 | 711 |
| NeuralLinear | 1009 | 549 | 1235 | 481 | 685 | 488 | 2853 | 291 |
| ConvNeuralLinear | 1160 | 540 | 2524 | 479 | 1643 | 596 | 2722 | 647 |
| Conv-Inf-NNGP | 29 | 8 | 322 | 181 | 21 | 7 | 157 | 42 |
| Conv-Inf-NTK | 49 | 32 | 425 | 322 | 28 | 7 | 907 | 711 |
| Conv-GP-NNGP | 15 | 2 | 221 | 118 | 37 | 5 | 830 | 533 |
| Conv-GP-NTK | 20 | 10 | 194 | 139 | 34 | 12 | 85 | 45 |
| Conv-Inf-NNGP-std | 17 | 3 | 732 | 432 | 66 | 15 | 889 | 442 |
| Conv-Inf-NTK-std | 52 | 31 | 99 | 64 | 8 | 0 | 27 | 12 |
| Conv-GP-NNGP-std | 20 | 7 | $^\dagger$101 | 59 | 100 | 55 | $^\dagger$124 | 49 |
| Conv-GP-NTK-std | 13 | 5 | $^\dagger$132 | 77 | 7 | 0 | $^\dagger$686 | 575 |
| Random | 141 | 61 | 402 | 184 | 471 | 398 | 485 | 395 |
| TuRBO-1 | 1150 | 638 | 4451 | 20 | 3865 | 289 | 4476 | 16 |
| TuRBO-5 | 3738 | 92 | 4456 | 37 | 4128 | 49 | 4466 | 26 |
| TPE | 1001 | 648 | 3901 | 140 | 3045 | 571 | 4119 | 156 |
| LIPO | 940 | 1073 | 1280 | 1073 | 1837 | 1792 | 2266 | 1626 |
| DIRECT-L | 20 | 0 | 4351 | 1 | 8 | 0 | 2525 | 38 |
| CMA | 9 | 1 | 4078 | 126 | 10 | 3 | 1777 | 969 |

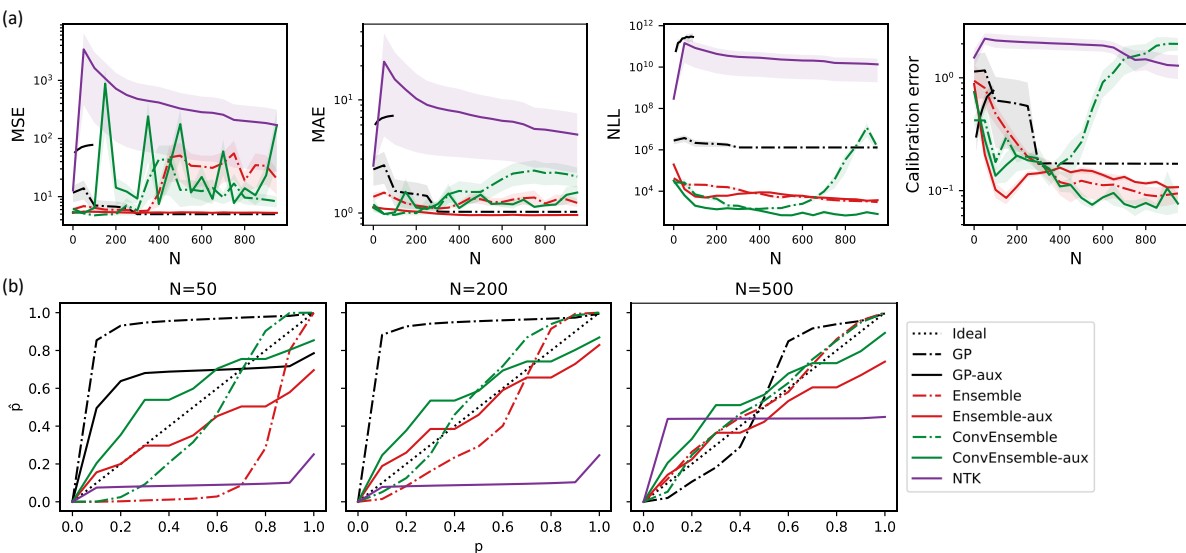

Figure 13: (a) Various metrics tracked during BO of the PC-A dataset distribution on a validation dataset of 1000 datapoints. (b) Uncertainty calibration curves measured at various points during BO. Note that the calibration curve for GP-AUX is only shown for $N = 50$, as it becomes computationally intractable for larger $N$.

**Validation Metrics**

To explore more deeply why certain surrogate models perform well while others do not, we track various metrics of the model during BO on a validation dataset with 1000 randomly sampled data points. In particular, we look at the mean squared error (MSE), the mean absolute error (MAE), the negative log-likelihood (NLL), and the calibration error on the PC-A data distribution. Results are shown in Figure 13(a).

The calibration error is a quantitative measure of the uncertainty of the model, which is important for the performance of BO as the acquisition function uses the uncertainty to balance exploration and exploitation. Intuitively, we expect that a 50% confidence interval contains the correct answer 50% of the time. In particular, we use the forecast calibration as proposed by Kuleshov et al. (2018), which is an extension of the calibration error proposed by Guo et al. (2017) to regression tasks:

$$\text{cal}(F_1, y_1, ..., F_T, y_T) = \sum_{j=1}^{m}(p_j - \hat{p}_j)^2$$

where $F_j$ is the CDF of the predictive distribution, $p_j$ is the confidence level, and $\hat{p}_j$ is the empirical frequency. We choose to measure the error along the confidence levels $p_j = (j-1)/10$ for $j = 1, 2, ..., 11$. The CDF $F_j(y_j)$ an be analytically calculated for models that have an analytical predictive distribution. For models that do not have an analytical predictive distribution, we use the empirical CDF:

$$F(y) = \frac{1}{n}\sum_{i=1}^{n}\mathbb{1}_{\mu^{(i)} \leq y}$$

where $\mathbb{1}$ is the indicator function. We also plot the calibration, $\{(p_j, \hat{p}_j)\}_{j=1}^{M}$, in Figure 13(b). Perfectly calibrated predictions correspond to a straight line.

Figure 13 shows that the infinite neural network kernel (NTK) has the highest prediction error, which is likely a contributing factor to its poor BO performance. Interestingly, vanilla GPs have the lowest MSE, so the prediction error is not the only indicator for BO performance. Looking at the calibration, the infinite neural network kernel has the highest calibration error, and we see from Figure 13(b) that it tends to be

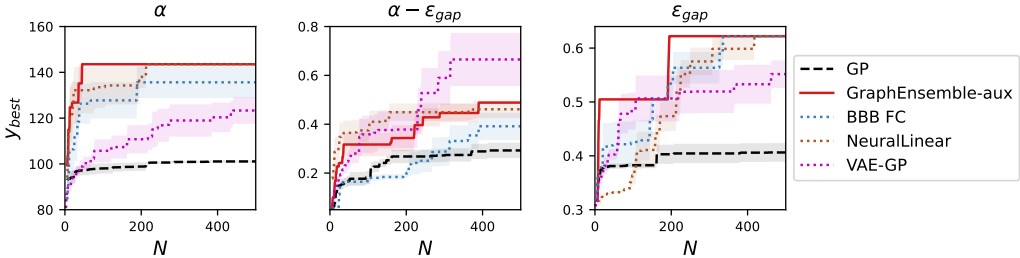

Figure 14: Additional BO results for several different objective functions on the chemistry dataset. GP and GRAPHENSEMBLE-AUX curves are replicated from the main text for convenience.

Table 6: BO results for the four different quantum chemistry objective functions. $^*$ denotes that $y_{\text{best}}$ is measured at $N = 100$ due to computational constraints.

|  | $y_{\text{BEST}}$ AT $N = 500$ | | | | | | | |
| MODEL | $\epsilon_{\text{GAP}}$ | | $-\epsilon_{\text{GAP}}$ | | $\alpha$ | | $\alpha - \epsilon_{\text{GAP}}$ | |
| | MEAN | SD | MEAN | SD | MEAN | SD | MEAN | SD |
|---|---|---|---|---|---|---|---|---|
| GP | 0.41 | 0.04 | $-0.10$ | 0.02 | 101.08 | 1.05 | 0.29 | 0.07 |
| GRAPHGP | $^*$0.62 | 0.00 | $^* -0.10$ | 0.02 | $^*$131.99 | 14.59 | $^*$0.24 | 0.03 |
| ENSEMBLE | 0.62 | 0.00 | $-0.08$ | 0.00 | 86.56 | 0.31 | 0.28 | 0.00 |
| ENSEMBLE-AUX | 0.62 | 0.00 | $-0.10$ | 0.02 | 83.86 | 4.45 | 0.13 | 0.05 |
| GRAPHENSEMBLE | 0.62 | 0.00 | $-0.10$ | 0.00 | 143.53 | 0.00 | 0.49 | 0.00 |
| GRAPHENSEMBLE-AUX | 0.62 | 0.00 | $-0.10$ | 0.00 | 143.53 | 0.00 | 0.49 | 0.00 |
| GRAPHBBB | 0.38 | 0.01 | $-0.11$ | 0.01 | 94.46 | 1.16 | 0.25 | 0.01 |
| GRAPHBBB-FC | 0.62 | 0.00 | $-0.10$ | 0.00 | 135.64 | 13.67 | 0.39 | 0.14 |
| GRAPHNEURALLINEAR | 0.62 | 0.00 | $-0.09$ | 0.01 | 143.53 | 0.00 | 0.46 | 0.09 |
| VAE-GP | 0.62 | 0.06 | - | - | 123.33 | 13.02 | 0.61 | 0.34 |
| VAE-GP-2 | - | - | - | - | 110.84 | 16.68 | 0.56 | 0.35 |
| VAE-GP-LATENT128 | - | - | - | - | 154.66 | 35.96 | 0.40 | 0.10 |
| VAE-GP-LATENT128-BETA0.001 | - | - | - | - | 133.66 | 13.25 | 0.42 | 0.13 |
| VAE-GP-LATENT32 | - | - | - | - | 114.83 | 14.64 | 0.53 | 0.38 |
| RANDOM | 0.38 | 0.02 | $-0.11$ | 0.03 | 105.19 | 7.87 | 0.29 | 0.07 |

overconfident in its predictions. GPs have a higher calibration error than the ensemble neural network methods, and tends to be significantly underconfident in its predictions. GP-AUX has higher validation loss, calibration error, and NLL than most, if not all, of the other methods, which explain its poor performance.

The ensemble NN methods tend to be reasonably well-calibrated. Within the ensemble NNs, the "-aux" methods have lower MSE and calibration error than their respective counterparts, and ConvEnsemble-aux has the lowest NLL calibration error out of all the methods, although interestingly Ensemble-aux seems to have the lowest MSE and MAE out of the ensemble NNs.

These results together show that calibration of Bayesian models is extremely important for use as surrogate models in BO.

### A.5.4 Organic Molecule Quantum Chemistry

The Bayesian graph neural networks (BGNNs) used for the chemical property optimization task consist of 4 edge-conditioned graph convolutional layers with 32 channels each, followed by a global average pooling operation, followed by 4 fully-connected hidden layers of 64 units each. The edge-conditioned graph convolutional layers Simonovsky & Komodakis (2017) are implemented by Spektral Grattarola & Alippi (2020).

More detailed results for the quantum chemistry dataset are shown in Table 6 and Figure 14. The architecture with the Bayes by Backprop variational approximation applied to every layer including the graph

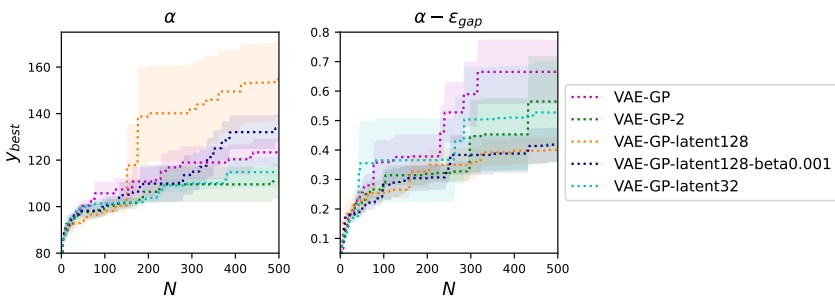

Figure 15: Additional BO results for VAE-GP using different pre-trained VAEs.

convolutional layers ("BBB"), performs extremely poorly, even worse than random sampling in some cases. However, only making the fully-connected layers Bayesian ("BBB-FC") performs surprisingly well.

Ensembles trained with auxiliary information ("ENSEMBLE-AUX") and neural linear ("NEURALLINEAR") perform the best on all objective functions. Adding auxiliary information to ensembles helps for the $\alpha$ objective function, and neither helps nor hurts for the other objective functions. Additionally, BNNs perform at least as well or significantly better than GPs in all cases. GPs perform comparably or worse than random sampling in several cases.

As noted in the main text, the performance of VAE-GP depends on the quality of the pre-trained VAE, as shown in Figure 15. The VAE-GP benchmark uses the same pre-trained VAE, and "VAE-GP-2" refers to the same method using a different random seed for the VAE. Even with the exact same method, VAE-GP-2 performs significantly worse on both objective functions. We also increase the latent space dimensionality from 52 to 128 in the "VAE-GP-LATENT128" benchmark, which performs even worse on the $\alpha - \epsilon_{\text{gap}}$ benchmark although it performs significantly better on the $\alpha$ benchmark. We also adjust the learning rate momentum to $\beta = 0.001$ in "VAE-GP-LATENT128-BETA0.001", and the latent space dimensionality to 32 in "VAE-GP-LATENT32". There is no clear trend with the different hyper-parameters, which may point to the random seed of the VAE pre-training being a greater factor in BO performance than the hyper-parameters.

**Validation Metrics**

As in Appendix A.5.3, we track the MSE, NLL, and calibration error during optimization on the chemistry task. Results are shown in Figure 16. The various metrics correlate with the respective methods' peformances during BO. For example, VAE-GP has an extremely high MSE and calibration error on the $\alpha$ objective, where it performs poorly, but has an MSE and calibration error more comparable with that of other methods as well as an extremely low NLL on the $\alpha - \epsilon_{\text{gap}}$ objective, where it performs extremely well. Likewise, the metrics for GRAPHGP are very high on the $\alpha - \epsilon_{\text{gap}}$ objective, where it performs poorly. GraphEnsemble tends to be among the better methods in terms of these metrics, which translates into good BO performance.

### A.5.5 Additional Discussion

BBB performs reasonably well and is competitive with or even better than ensembles on some tasks, but it requires significant hyperparameter tuning. The tendency of variational methods such as BBB to underestimate uncertainty is likely detrimental to their performance in BO. Additionally, Sun et al. (2019) shows that BBB has trouble scaling to larger network sizes, which may make them unsuitable for more complex tasks such as those in our work. BOHAMIANN performs very well on the nanoparticle narrowband objective and comparable to other BNNs without auxiliary information on the nanoparticle highpass objective. This is likely due to its effectiveness in exploring a multi-modal posterior. However, the need for SGHMC to sample the posterior makes this method computationally expensive, and so we were only able to run it for a limited number of iterations using a small neural network architecture.

Infinitely wide neural networks are another interesting research direction, as the ability to derive infinitely wide versions of various neural network architectures such as convolutions, and more recently graph con-

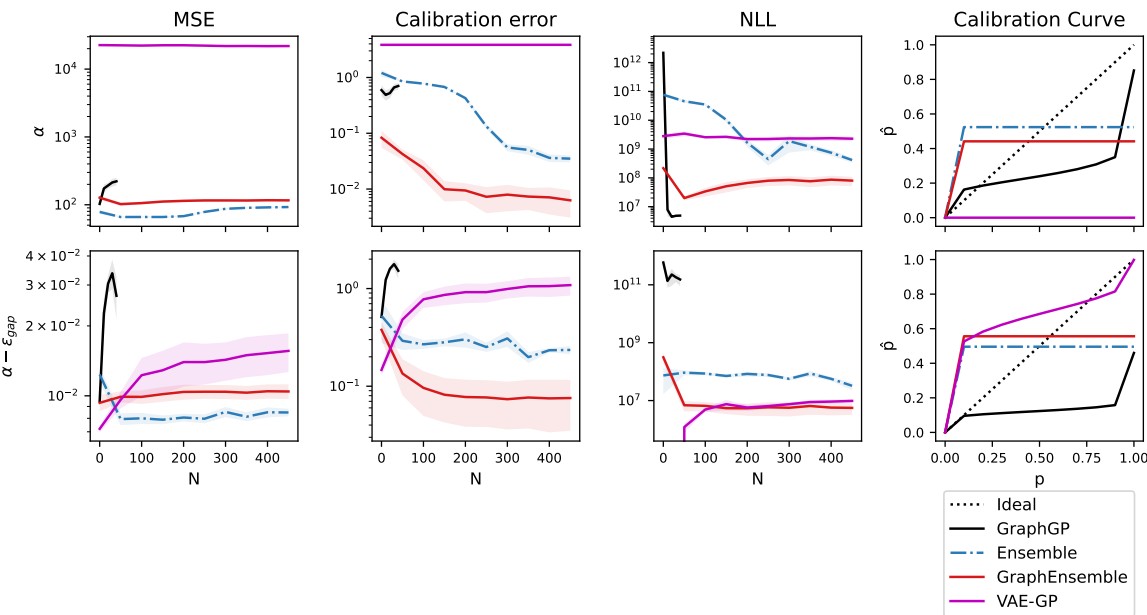

Figure 16: (a) Various metrics tracked during BO of the PC-A dataset distribution on a validation dataset of 1000 datapoints. (b) Uncertainty calibration curves measured at various points during BO.

volutional layers (Hu et al., 2020) could potentially bring the power of GPs and BO to complex problems in low-data regimes. However, we find they perform relatively poorly in BO, are quite sensitive to hyper-parameters (e.g. kernel and parameterization), and current implementations of certain operations such as pooling are too slow for practical use in an iterative setting. In particular, BO using an infinite ensemble of infinite-width networks performs poorly compared to normal ensembles, suggesting that the infinite-width formulations do not fully capture the dynamics of their finite-width counterparts.

Non-Bayesian global optimization methods such as LIPO and DIRECT-L are quite powerful in spite of their small computational overhead and can even outperform BO on some simpler tasks. However, they are not as consistent as BO, performing more comparably to random sampling on other tasks. CMA-ES performs poorly on all the tasks here. Also, like GPs, these non-Bayesian algorithms assume a continuous input space and cannot be effectively applied to structured, high-dimensional problems.

### A.6 Compute

All experiments were carried out on systems with NVIDIA Volta V100 GPUs and Intel Xeon Gold 6248 CPUs. All training and inference using neural network-based models, graph kernels, and infinite-width neural network approximations are carried out on the GPUs. All other models are carried out on the CPUs.

