# OpenReview forum: "Deep Learning for Bayesian Optimization of Scientific Problems with High-Dimensional Structure"
_TMLR — Accepted by TMLR_

### Review · Reviewer_Trh3 · 2022-08-01

**Summary Of Contributions:**

The paper studies the problem of optimizing expensive black-box functions in high dimensions with BO framework when we have additional auxiliary information for e.g. a-priori known structure of the input domain (e.g. symmetries or invariances). The key idea is to employ Bayesian neural networks (BNNs) surrogate models with model architectures capturing the auxiliary information for e.g. like convolutional NNs to exploit translation invariance in images, or graph NNs to exploit symmetries in molecule.

**Requested Changes:**

Overall, I think the paper is at a workshop level right now. However, adding new insights and new qualitative/quantitative discussion with respect to a large body of existing work (as mentioned in the review) will be useful and strengthen the paper.

**Strengths And Weaknesses:**

- The problems in the experimental section are interesting (catering to scientific applications) and will be useful for the community if released (currently the attached repo is empty https://github.com/placeholder/deepBO)
- In terms of methodology, unfortunately there is limited new knowledge in the paper. BNNs have been used earlier in multiple papers for BO and there doesn't seem to be any new technique to train them introduced here. One point about using cosine annealing learning rate (Loshchilov and Hutter) is mentioned when retraining the model on acquiring a new point instead of re-training from scratch. However, Figure 7 shows that the gains from this technique are not very effective.
- One other big concern is about the baselines. There is no comparison/discussion with any high-dimensional BO approach using GPs like TurBO, SAASBO, BOCK etc. TurBO can easily be scaled to 10000 points much larger than the sizes in the experiment of the paper (1000).

TurBO: "Scalable global optimization via local bayesian optimization." Advances in Neural Information Processing Systems 32 (2019)

SAASBO: David Eriksson and Martin Jankowiak. High-dimensional bayesian optimization with sparse axis- aligned subspaces. In Uncertainty in Artificial Intelligence, pages 493–503. PMLR, 202

BOCK: ChangYong Oh, Efstratios Gavves, and Max Welling. Bock: Bayesian optimization with cylindrical kernels. In International Conference on Machine Learning, pages 3868–3877. PMLR, 2018

There is no comparison with any of the latent space VAE style approaches as well.

Local Latent Space Bayesian Optimization over Structured Inputs. CoRR abs/2201.11872 (2022)

Combining Latent Space and Structured Kernels for Bayesian Optimization over Combinatorial Spaces. NeurIPS 2021: 8185-8200

Sample-Efficient Optimization in the Latent Space of Deep Generative Models via Weighted Retraining. NeurIPS 2020

---

> ### Author Response · Authors · 2022-08-05
> **Response to Trh3**
>
> Thank you very much for your detailed review! I have responded to specific comments below.
>
> > currently the attached repo is empty
>
> Thank you for pointing this out, I have posted an anonymized version of our code to this repo if you are interested: https://github.com/throwaway314/DeepBO
>
> > One other big concern is about the baselines. …
>
> Thank you for suggesting these works. I will expand our discussion to include these high-dimensional BO approaches. However, these approaches are not comparable baselines as we are studying a different type of high-dimensionality. As you pointed out, TurBO focuses on scaling BO to large number of evaluations, which is not an explicit focus of our work. In the moderate-data regime (N=1000), vanilla BO with GPs are still computationally tractable to make a comparison. It could be interesting to combine TurBO with some of the more expensive GP methods we compared to (convGPs and graphGPs) although that would be outside the scope of this work.
>
> Additionally, works such as SAASBO and BOCK focus on tackling continuous high-dimensional input spaces by either assuming a structure (e.g. additive structure) or assuming a low-dimensional subspace (e.g. through a cylindrical transformation). Our work however focuses on the setting of non-continuous high-dimensional input spaces, i.e. images and graphs, that require the use of specialized kernels or neural network architectures.
>
> > There is no comparison with any of the latent space VAE style…
>
> While the results are not in the main text, we have actually already included a comparison to latent-space methods. In particular, we used the code from Tripp et al. NeurIPS 2020, which we labeled as “VAE-GP”. We give a model description in Section 3 and briefly discuss the results in Section 5. More detailed results can be found in Appendix A.8.

---

### Review · Reviewer_Cgt7 · 2022-08-09

**Summary Of Contributions:**

This paper studies Bayesian optimization using Bayesian neural networks and ensembles of neural networks as the surrogate models for optimizing three complicated scientific design tasks (multilayer nanoparticles, crystal topology, and quantum chemistry). The authors also demonstrate the success of using auxiliary information, e.g. information outside the existing objective, as well as the strong performance of simple ensembles of neural networks compared to simple Gaussian process models.

The authors use both cyclical learning rates for continued retraining of their networks (an advance over BOHIMIANN, which instead uses scale adaptation, a weaker type of retraining) as well as incorporate auxiliary information (like image outputs from the simulators) into their NNs (much like Astudillo & Frazier, ’19 and Balandat et al, ’20 did with GPs).
Overall, the authors demonstrate the importance of high performing surrogate models, presumably in terms of accuracy, for having highly performant Bayesian optimization loops for scientific design problems.

**Would some individuals in TMLR's audience be interested in the findings of this paper?**

Yes, Bayesian optimization for scientific design and discovery is an extremely interesting set of applications for machine learning. The findings in this paper should certainly interest people who read TMLR.

**Are the claims made in the submission supported by accurate, convincing and clear evidence?**

Possibly, I still think that the interplay between Bayesian neural nets, ensembles, and Gaussian processes could be better studied and understood in this work. Please see the following sections.



**Requested Changes:**

_Model fits (medium / high):_

I think that showing model fits, describing the performance on a heldout set on each task (as much as possible), for a couple of surrogate models would vastly clean up the story of this paper. Judging from the experiments, it’s sort of easy to tell when various models are likely performing well in prediction tasks (e.g. the Graph nets on Fig. 5 or the GP on highpass in Fig. 2), but it would be really nice to tell this type of performance for sure.

After all, the experimental results seem to demonstrate to me that one should use the best performing surrogate model for a given problem (as long as it has reasonable uncertainty – show the ensembles have reasonable uncertainty please) and be somewhat agnostic about using GPs or any other model class. However, this type of message would be a lot more clear with these types of model fits.
This would also be helpful in diagnosing the underperformance of the GP-aux model as well (Fig. 4). A similar argument could be made to study the weak performance of the infinite NNs throughout (less flexibility produces worse performance).

It also would help parse out when / how auxiliary information helps – does it help in the objective itself, or just in uncertainty. How exactly?

Code (medium / high): Providing the code would be helpful. I’d personally like to see the implementation of the graphGP, as it seems very strange that it both uses the GPU and is somehow 15x slower than a GP on the CPU.

_Comparison with TurBO (medium):_

I still think that a strong baseline, especially in these sorts of large n problems would be helpful.

-	Useful reference is Eriksson et al, ’19 and the implementations in BoTorch (https://botorch.org/tutorials/turbo_1) as well as LaMCTS (Wang et al, ’20) (https://github.com/facebookresearch/LaMCTS/) which certainly scales even farther.

-	GP-EI is a pretty weak baseline for a GP enabled BayesOpt loop these days (UCB, KG, MVES, etc..), given the vast amount of BayesOpt tooling that exists, particularly from BoTorch.

_More detailed study of VAE-GP (medium):_

 Fig. 13 is really interesting and probably deserves space in the main text given the popularity of deep models + GPs for BayesOpt. Perhaps the authors could cut down on sum of the ending discussion in order to move it into the main text and to flesh out the experiment a bit more, with a bit more detail there.

_Claims (medium):_

“Neural Linear and GPs scale cubically with output dimensionality (without the use of covariance approximations),”
-	This claim is still a bit misleading. Various multi-task Gaussian processes aren’t exactly “approximate GPs” or even “covariance approximations”, they’re different model classes than a GP fit over all nt data points, that make different independence assumptions than standard GPs. All covariances are exact in a MTGP implementation, see Bonilla et al, ’08 and also Maddox et al, ’21 for a BayesOpt application.

Reference Wang et al, ’18 is all about scaling BayesOpt to tens of thousands of data points, not just pure prediction, and so probably deserves a bit more detail / comparison to your approaches. Even if, in my understanding, it is quite computationally intensive.

Does performance depend on ensemble size?

_Small fixes:_

Section 1.1: don’t use “Ref”. Strike those words.

Fig 13: the comparison with the VAE-GP still probably needs a bit more comparison and explanation. Why is the VAE precisely so helpful on only one task, which seems to be the most difficult to model?
-	Also, why no error bars on the GraphEnsemble (also seems to apply to Fig. 5)?

I feel like I’m missing this somewhere, but how many trials were each experiment run over?


Space after graphs at second to last paragraph of Section 1.

_References:_

Eriksson, David, et al. "Scalable global optimization via local bayesian optimization." Advances in Neural Information Processing Systems 32 (2019).

Wang, Linnan, Rodrigo Fonseca, and Yuandong Tian. "Learning search space partition for black-box optimization using monte carlo tree search." Advances in Neural Information Processing Systems 33 (2020): 19511-19522.

Bonilla, Edwin V., Kian Chai, and Christopher Williams. "Multi-task Gaussian process prediction." Advances in neural information processing systems 20 (2007).

Maddox, Wesley J., et al. "Bayesian optimization with high-dimensional outputs." Advances in Neural Information Processing Systems 34 (2021).



**Strengths And Weaknesses:**

As a full disclosure, I’ve reviewed this work previously, and do appreciate many of the changes made. The figures are of much higher quality now, and the study on VAE – GPs is much clearer now.

**Strengths:**

The description of the tasks and their relevance to a scientific audience is a selling point. I also appreciate the charts of their descriptions.

Experimental design is generally quite broad and incorporates good baselines:
-	Experiments using CMA-ES (and other non BO approaches) (Fig. 9), even if LIPO tends to perform quite well.

-	Experiments on tabular data, even if standard BO performs quite well (Fig. 8).

-	Knockout / ablation experiments on several of the test problems. (Figs. 6, 10).


**Weaknesses:**

The scientific and experimental claims are still a bit confusing to me (see Model fits, TuRBO comparison, and VAE-GP proposals for fixing).

-	The abstract + intro could be re-worked a bit more to make the claims / experiments in the main text stand out. For example, VAE-GPs are mentioned, but they don’t appear anywhere in the main text.

-	The takeaways from the paper proposals could be editorialized a bit more – what exactly did you demonstrate (that inductive biases and NN ensembles help in BayesOpt, e.g. well designed surrogate models)?

Code isn’t provided (but I do hope that this would be remedied on acceptance).

---

> ### Author Response · Authors · 2022-08-09
> **Code**
>
> Thank you so much for re-reviewing our work, and for the detailed feedback. We will try to address as many of these as possible.
>
> > Code isn’t provided (but I do hope that this would be remedied on acceptance).
>
> I forgot to update the link in the manuscript upon submission, but I have posted an anonymized version of our code to this repo if you are interested: https://github.com/throwaway314/DeepBO. Regarding the graphGP, we are using the WL kernel implementation provided by Ru et al (2021), the code for which is copy + pasted inside the `nasbowl/` directory.

---

> > ### Comment · Reviewer_Cgt7 · 2022-09-06
> > **Thanks**
> >
> > Thank you for posting the code.

---

> ### Author Response · Authors · 2022-08-18
> **Detailed response to Reviewer Cgt7**
>
> Thank you again for your detailed review. We have been running experiments, and now that results are starting to come in, we wanted to address your review in more detail.
>
> > Model fits (medium / high):
>
> Thank you for your recommendation and for your thoughts on why certain models work better than others. We have re-run the Bayesian optimization on the photonic crystal dataset, tracking various metrics on a validation dataset along the way. As you suspected, the infinite NN kernel has the largest MSE, which is likely a contributing factor to its poor performance in Bayesian optimization. However, vanilla GPs have the lowest validation MSE, so the story is not complete.
>
> We also tracked the negative log-likelihood (NLL) (e.g. as formulated by Eq. 1 in Lakshminarayanan et al.) and the calibration error (as proposed by Kuleshov et al. which was an extension of the work by Guo et al to regression tasks). The calibration error is the largest for the infinite-NN kernel by a large margin, and the empirical frequency (Eq. 8 of Kuleshov et al) shows that this method is significantly overconfident in its predictions. GPs on the other hand are significantly underconfident in their predictions. The ensemble NN methods tend to be reasonably well-calibrated. The NLL can be interpreted as a metric combining the MSE and the uncertainty quality, and shows similar conclusions in that the NLL of infinite NN kernel is orders of magnitude larger than that of the GP, which in turn is significantly larger than that of the ensemble NN methods. Within the ensemble NNs, the "-aux" methods have lower MSE and NLL than their respective counterparts, and ConvEnsemble-aux has the lowest NLL and calibration error out of all the methods.
>
> GP-aux is unfortunately still running, but we will update the paper with these results as well.
>
> > Comparison with TurBO (medium):
>
> We have run two settings of TuRBO (m=1 and m=5 where m is the number of trust regions) on the nanoparticle scattering tasks. Both variants perform similarly to GPs on the highpass objective (slower to converge, but find a better maxima) whereas TuRBO-1 does not perform as well on the narrowband objective. We will include these baselines in the manuscript as well as run experiments on the photonic crystal dataset.
>
> > GP-EI is a pretty weak baseline for a GP enabled BayesOpt loop these days
>
> Thank you for this recommendation. We chose EI as it was one of the more popular choices at the time, and it lends itself very well to Monte Carlo methods for optimizing the inner loop, which is necessary for using BNNs as surrogates (naive TS is computationally expensive with ensembles, and UCB can result in ties). We used EI for GPs to maintain consistency. TuRBO uses TS which provides an interesting point of comparison, and it would be interesting to consider extending TS, KG, etc to BNNs in future work.
>
> > Fig. 13 is really interesting and probably deserves space in the main text
>
> Thank you for this suggestion, we will try to move this into the main text.
>
> > “Neural Linear and GPs scale cubically with output dimensionality (without the use of covariance approximations),” - This claim is still a bit misleading.
>
> I might be misunderstanding, but to my knowledge, MTGPs still scale cubically with the output dimensionality. Bonilla et al state that standard MTGPs take $\mathcal{O}(n^3t^3)$, which they are able to push down to $\mathcal{O}(ntp^2q^2)$ through the use of approximations. Table 1 in Maddox et al shows that they can get the complexity down to $\mathcal{O}(n^3+t^3)$ using Matheron's rule on the outputs.
>
> > Reference Wang et al, ’18 is all about scaling BayesOpt
>
> Thank you for pointing this out, we will expand the discussion on this.
>
> > Does performance depend on ensemble size?
>
> We have not tested the ensemble size rigorously, although some preliminary results showed that performance does not increase above size > 10.
>
> > I feel like I’m missing this somewhere, but how many trials were each experiment run over?
>
> Experiments were run over 5 trials. Thank you for pointing this out, we will include it in method details.
>
> > Fig 13: the comparison with the VAE-GP still probably needs a bit more comparison and explanation.
>
> It is unclear to us why it performs well on certain metrics, and we plan to add experiments tracking the MSE/calibration/NLL on the chemistry task.
>
> ## References:
>
> Guo, C., Pleiss, G., Sun, Y., & Weinberger, K. Q. (2017, July). On calibration of modern neural networks. In International conference on machine learning. PMLR.
>
> Kuleshov, V., Fenner, N., & Ermon, S. (2018, July). Accurate uncertainties for deep learning using calibrated regression. In International conference on machine learning. PMLR.
>
> Lakshminarayanan, B., Pritzel, A., & Blundell, C. (2017). Simple and scalable predictive uncertainty estimation using deep ensembles. Advances in neural information processing systems.

---

### Review · Reviewer_bRGJ · 2022-08-12

**Summary Of Contributions:**

The paper discusses three different scientific optimization problems with high-dimensional search spaces form physics and chemistry applications, that exhibit known structures or auxiliary information and how to solve them with Bayesian optimization. Due to the high dimensionality of the data, the paper proposes deep neural networks as probabilistic model for Bayesian optimization, which scale better with the number of data points and dimensions than more commonly used Gaussian processes.

**Broader Impact Concerns:**

I do not see any ethical concerns with this work

**Requested Changes:**

Section 4.3: It is unclear how much the graph neural network actually helps for this task compared to using a ensemble based on MLPs that operates on the SOAP descriptor directly. It would be great if the authors could add this as additional baseline

Please also add a comparison to Bayesian optimization with other probabilistic models such as random forests or kernel density estimators. Furthermore, it would be nice if the VAE-GP baseline could be also applied for the other two tasks.


## Minor comments:

- Typo Section 1.1 drop the word  'Ref' in the second and third sentence.

- Can you increase the figure size to make the plots more visible? You waste a lot of white space for the margins. Alternatively, you can move some plots from the appendix to the main paper, to fill these white spaces.

- Figure 5: Does 'GraphEnsemble' in the subplot (c) corresponds to BGNN? If so, could you use the same label?



**Strengths And Weaknesses:**

## Strengths

### Main claims are supported by experiments

The first claim of the paper is that auxiliary information helps to speed-up the optimization process. On all three benchmarks the proposed BO strategy with neural networks ensembles achieves the same or faster convergence when it is augmented with auxiliary information. The only exception is for the Organic Molecule Quantum Chemistry with a HOMO-LUMO energy gap, where the GraphEnsemble seems to perform slightly worse when augmented with auxiliary performance.

Second, the paper claims that BO with convolutional or graph neural networks achieves a better performance on complex input spaces such as images or molecules. This is shown for the Photonic Crystal Topology benchmark. However, as mentioned below, I miss another baseline to support this claim for the Organic Molecule Quantum Chemistry that uses standard feed forward neural networks instead of graph neural networks.

Lastly, the paper claims that neural networks outperform Gaussian processes on these problems. This is also supported by empirical evidence at least for two benchmarks. However, on the Multilayer Nanoparticle GPs without auxiliary data seem to work slightly better than the proposed BO strategy with neural networks.


### Relevance to the community

I think the paper could be interesting for the BO community because it a) provides some interesting and challenging benchmarks that encourage further research into this direction and b) establishes some sensible baselines for neural network based Bayesian optimization.

## Weaknesses


### Empirical evaluation can be improved

Apart from adding the missing baseline for the Organic Molecule Quantum Chemistry, I think the empirical evaluation of the paper can be further improved by a comparison to other Bayesian optimization method that use other probabilistic models, such as random forests or kernel density estimators, which have shown competitive performance on high dimensional structured spaces before. Also, I don't understand why the VAE based GP method has only be considered for the Organic Molecule Quantum Chemistry - where it achieved competitive performance -  and not for the other two tasks.


### Further clarifications

Some parts of the paper require further clarifications:

- Equation 3 seems wrong. Normally you first compute the empirical mean \mu = \sum_i \mu^(i) and variance \sigma = \sum_i (\mu^(i) - \mu)^2) of your ensemble and use \mu and \sigma to compute Equation 2.

- Section 2.2:  How exactly do you use cosine annealing here? For how many steps? I also don't understand how cosine annealing here helps to learn the new data point? Do you also use consine annealing if you train from scratch?

- How do you optimize the acquisition function for neural network based Bayesian optimization methods?

- Figure 5 (d): Do you train both GraphGP and GraphEnsemble on GPU or CPU?

---

### Decision · Action_Editors · 2022-09-21

**Recommendation:** Accept as is

**Comment:**

This paper demonstrates through numerical examples that Bayesian Optimization (BO) can be applied in relatively high dimensional settings when it is augmented with auxiliary information through Bayesian neural networks surrogate models.

The work advances the understanding of which methods work well in which settings, provides some interesting and challenging benchmarks for BO and establishes sensible baselines for neural network-based Bayesian optimization.

Three reviewers provided detailed feedback on the manuscript, engaged with the authors during the rebuttal and unanimously stated that the paper has become much stronger after the revisions. The reviewers concluded that the acceptance criteria for TMLR are met, and all reviewers recommend acceptance.

Since the reviewer argued that the numerical benchmarks could serve as a reference for future work, I would recommend adding a short table to the camera-ready version that summarizes the key features (such as dimensionality) of the benchmark problems to make this information more readily available to the readers.